# CRISPR-Cas9 System for Plant Genome Editing: Current Approaches and Emerging Developments

**Jake Adolf V. Montecillo [†], Luan Luong Chu [†] and Hanhong Bae ***

Department of Biotechnology, Yeungnam University, Gyeongsan, Gyeongbuk 38541, Korea;
adolfjake@ynu.ac.kr (J.A.V.M.); chull@yu.ac.kr (L.L.C.)
* Correspondence: hanhongbae@ynu.ac.kr; Tel.: +82-53-810-3031
† These authors contributed equally to this work.

**Abstract:** Targeted genome editing using CRISPR-Cas9 has been widely adopted as a genetic engineering tool in various biological systems. This editing technology has been in the limelight due to its simplicity and versatility compared to other previously known genome editing platforms. Several modifications of this editing system have been established for adoption in a variety of plants, as well as for its improved efficiency and portability, bringing new opportunities for the development of transgene-free improved varieties of economically important crops. This review presents an overview of CRISPR-Cas9 and its application in plant genome editing. A catalog of the current and emerging approaches for the implementation of the system in plants is also presented with details on the existing gaps and limitations. Strategies for the establishment of the CRISPR-Cas9 molecular construct such as the selection of sgRNAs, PAM compatibility, choice of promoters, vector architecture, and multiplexing approaches are emphasized. Progress in the delivery and transgene detection methods, together with optimization approaches for improved on-target efficiency are also detailed in this review. The information laid out here will provide options useful for the effective and efficient exploitation of the system for plant genome editing and will serve as a baseline for further developments of the system. Future combinations and fine-tuning of the known parameters or factors that contribute to the editing efficiency, fidelity, and portability of CRISPR-Cas9 will indeed open avenues for new technological advancements of the system for targeted gene editing in plants.

**Keywords:** CRISPR-Cas9; genome editing; transgene-free; off-target effect; ribonucleoproteins (RNPs); multiplexing; viral vectors; heat stress

## 1. Introduction

Genetic diversity plays a crucial role in the development of novel plant varieties. In this regard, gene diversification for the improved genetic architecture of agricultural crops has been practiced for years via conventional plant breeding techniques or through physicochemical and biological-induced random mutagenesis [1]. Although these techniques have been successfully utilized over the ages and have augmented crop production, modern approaches exploiting the promises of genome editing offer a precise and rapid development of improved plant varieties [2].

Genome editing is a collection of strategies and techniques developed to make a defined or tailored alterations in the genetic composition of an organism [3]. The key tool of this editing technology is the use of site-specific nuclease (SSN), a programmable nuclease that targets specific gene sequences in a precise manner. The use of these engineered nucleases to precisely delete, insert, and replace specific gene sequence highlights the edge of the site-directed mutagenesis approach and thus is favored over random mutagenesis techniques [4]. Genome editing technologies include customized homing nuclease (meganuclease), zinc-finger nucleases (ZFNs), transcription activator-like effector nucleases

(TALENs), which all rely on protein-based systems with customizable DNA-binding specificities for targeting gene sequence, and the more recent platform clustered regularly interspaced short palindromic repeats-CRISPR associated 9 nuclease (CRISPR-Cas9), which depends on RNA as a targeting moiety that directs the nuclease to a defined DNA sequence [5,6]. These genome editing technologies work through employing site-directed nucleases to induce double-strand breaks (DSBs) at predefined genomic loci. The formation of DSBs stimulates the recruitment of endogenous host DNA repair factors that act in either of the two repair pathways: homologous recombination (HR) or non-homologous end-joining (NHEJ) [7,8] (Figure 1). HR repairs DSB through the integration of sequences that contain sequence homology flanking the DSB. HR-mediated repair thus can be used to insert desired sequences through recombination of exogenous repair or donor templates with the target locus [9]. DSB can be also repaired via the error-prone NHEJ machinery that introduces random insertion or deletion mutations (indels) in varying lengths, resulting in frameshift mutations in the coding sequence, thereby creating gene knockouts [1]. In contrast to the HR-mediated repair mechanism, the NHEJ repair pathway occurs with higher frequency in most organisms, including plants [10]. The ability of these genome editing technologies to induce DSB at specific genomic locus has made them a powerful tool for targeted genome modifications.

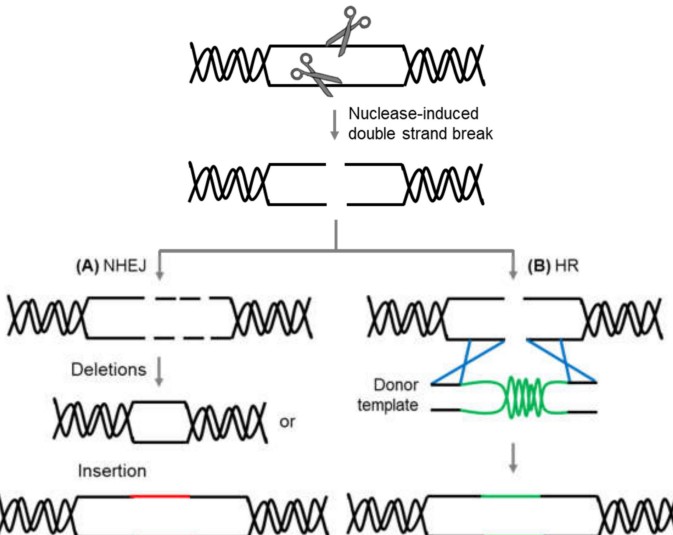

**Figure 1.** Site-specific nuclease-mediated genome editing. Nuclease-induced double-strand breaks (DSBs) can be repaired by host endogenous repair pathways: non-homologous end joining (NHEJ) and homologous recombination (HR). (**A**) NHEJ can introduce nucleotide deletions or insertions (indels) at the site of DSB during the repair process, resulting in gene mutation or knock-out. (**B**) With the presence of donor template bearing homologous sequence flanking the site of DSB, HR-mediated repair is achieved leading to precise integration (knock-in) of the specific gene sequence.

The advent of genome editing has drastically influenced the progress of plant biology, agriculture, and other scientific fields as it allows precise genetic alterations of model systems [11]. Several studies have demonstrated the efficacy of these genome editing approaches in modifying plant genome [12–15]. Of the existing genome editing technologies, the recently adopted CRISPR-Cas9 genome editing platform has drawn wide attention from the scientific community. CRISPR-Cas9 system has shown great advantages and is more straightforward compared to the early developed methods for targeted genome editing (meganuclease, ZFNs, and TALENs). Unlike the other genome editing technologies that require a time-consuming and resource-intensive protein engineering process to create synthetic DNA-binding proteins, the CRISPR-Cas9 system provides a simpler, less expensive, and more versatile approach based on RNA-guided nucleases for genome editing with higher success rates [7,16,17]. The application of CRISPR-Cas9 technology in model plant systems to elucidate and manipulate

gene functions has generated substantial knowledge about the accuracy and efficiency of its genome editing ability [18]. CRISPR-Cas9-mediated genome modification thus far has been performed not only in model plant systems but also in many economically important crops, producing high quality and sustainable products [1]. In plants, however, the success of the CRISPR-Cas9 system to induce site-directed mutagenesis relies on the efficiency of suitable vector and other molecular components (Cas9 gene sequence, promoters, and guide RNA design), transformation/delivery methods and the ability of the plant to regenerate [19]. Proper consideration and planning are thus needed in implementing this genome editing system.

This review provides a concrete overview of the fundamentals of the CRISPR-Cas9 system and its application in plant genome editing. Details on the existing challenges, limitations, and practical considerations in implementing the system are also discussed. The review also presents the current and emerging methodological advancements and optimization approaches for simple and efficient genome editing systems.

## 2. CRISPR-Cas: Defense to Editing System

Bacteria are in a persistent battle with bacteria-specific viral predators—bacteriophages. To fend off bacteriophage infection, bacteria have evolved diverse immune mechanisms that act at different stages of the phage infection process [20]. These immune mechanisms described so far involve phage attachment aversion, DNA entry blockage, restriction-modification systems, abortive infection, assembly interference, and CRISPR-Cas system. Among these immune mechanisms, only the CRISPR-Cas system represents the known adaptive immune system in prokaryotes [20,21]. The mechanism of CRISPR-Cas immunity is divided into three distinct stages: integration of short sequences as "spacers" from invading foreign DNA, biogenesis of CRISPR RNA (crRNA), and interference of target DNA by crRNA [22]. In the first stage, the acquisition of a new spacer sequence happened through the processing of the identified foreign DNA and integration into the CRISPR locus. The CRISPR locus is then transcribed in the next stage to generate a single RNA, which is subsequently processed to form shorter mature crRNAs, each containing a single spacer sequence. During the final stage, an effector complex is formed comprising of a Cas nuclease and crRNA. The effector complex then destroys the foreign DNA that carries a sequence complementary to the crRNA [23] (Figure 2). The immunity conferred by the CRISPR-Cas system relies on the library of spacer sequences matching a phage genome or foreign DNA. The infection of a novel phage leads to the addition of new spacer sequences, resulting in the expansion of the CRISPR array [24]. CRISPR-Cas system keeps records of past infections as CRISPR array, allowing it to elicit rapid immune response upon reinfection process [23].

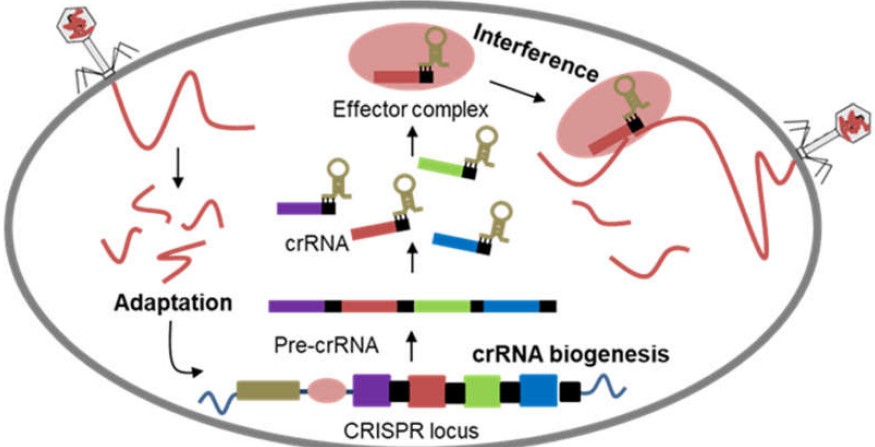

**Figure 2.** Mechanism of CRISPR-Cas system-mediated immunity. The immunity conferred by the CRISPR-Cas system occurs in three successive stages. The introduction of foreign DNA during bacteriophage

infection triggers the activation of the adaptation machinery, of which the invading DNA is processed into protospacers and incorporated into the CRISPR locus. Transcription of the CRISPR locus leads to the biogenesis of crRNAs, formed from the processing of the initial pre-crRNA transcript. In the interference stage, the crRNA assembles with Cas nuclease to form an effector complex that recognizes and degrades foreign DNA upon subsequent infection.

The discovery of the CRISPR-Cas system originated from the identification of unusual structures downstream of the *iap* gene in *Escherichia coli*. The structure is composed of five homologous sequences of 29 nucleotides (nt) arranged in direct repeats separated by 32 nt as spacers [25]. These interspaced repeat sequences were later on referred to as CRISPR, with associated *cas* genes [26]. CRISPR-Cas system is encoded by a genomic locus consisting of a series of direct repeat sequences separated by variable spacer sequences known as a CRISPR array, and a diversity of flanking *cas* genes [26,27]. Among microbial genomes, CRISPR-Cas systems show remarkable diversity and are currently classified into two distinct classes: Class 1 and Class 2. The classification scheme is based on the combination of several criteria: the signature *cas* genes in each CRISPR-Cas locus, the phylogeny of the conserved Cas protein—Cas1, and the arrangement of genes in the CRISPR-Cas loci [28,29]. These two classes of CRISPR-Cas system differ mainly in their effector modules. Class 1 CRISPR-Cas systems are defined by having effector complexes consisting of multiple subunits of Cas proteins. Under this class are types I, III, and IV CRISPR-Cas systems. In contrast to the class 1 system, class 2 has a simpler architecture of the effector module comprising a single protein with multiple domains and functions. This class includes types II, V, and VI CRISPR-Cas systems [28,30,31]. Details of this classification have been thoroughly discussed in the recent review of Makarova et al. [32]. Class 2 systems have become an attractive avenue to devise genome editing platforms, attributing to its relatively simple organization of effector complexes. In particular, the type II CRISPR-Cas systems with their Cas9 endonuclease have been widely exploited today as tools for genome editing in a variety of organisms [33].

Considering the stages of CRISPR-Cas immunity, the type II CRISPR-Cas systems integrate short sequences from invading foreign DNA into the CRISPR array in the host genome [24]. Transcription of the CRISPR array results in the formation of a single pre-crRNA transcript that is subsequently processed to form mature short crRNAs that can pair with complementary sequences (protospacers) of the invading foreign DNA. This process of pre-crRNA maturation is directed by the hybridization of another RNA known as the trans-activating crRNA (tracrRNA) and facilitated by the activities of RNase III and Cas9 protein [34]. The crRNA and tracrRNA base-paired structure forms a complex with the Cas9 nuclease, activating Cas9 for site-targeted cleavage of DNA. Cleavage of the target DNA by Cas9 nuclease occurs at sites determined by the crRNA with a complementary sequence to the target protospacer DNA lying adjacent to a short sequence referred to as protospacer adjacent motifs (PAMs). The PAM is an essential recognition sequence that determines the target strand for cleavage. It also serves as a motif that delineates CRISPR loci from other genetic material, preventing the loci itself as the target. Different types of CRISPR-Cas systems require different PAM sequences [35,36]. In general, type II CRISPR-Cas systems require three components (crRNA, tracrRNA, and Cas9 nuclease) to target and induce cleavage of DNA (Figure 3A).

The extensively characterized type II CRISPR-Cas system of *Streptococcus pyogenes* has been widely adopted today as a platform for genome editing in many biological systems. This genome editing system from *S. pyogenes* has been engineered to improve the ease of implementing genome modifications. The dual crRNA-tracrRNA has been fused as a single RNA chimera known as the single guide RNA (sgRNA) (Figure 3B). This has simplified the system from the typical three components into two components: sgRNA and Cas9 nuclease, which must be introduced or expressed in an organism to carry out genome editing. The sgRNA is usually composed of 20 nt at the 5′ end corresponding to the protospacer sequence and will direct the Cas9 nuclease to the target DNA site through RNA-DNA base-pairing rules. An appropriate PAM sequence must be present at the 3′ end of the protospacer sequence for cleavage to occur. In *S. pyogenes* system, the canonical PAM sequence is 5′-NGG, where N could be any nucleotide [35,36]. With this engineered system, genome editing can be performed by

simply altering the 20 nt sequences of the sgRNA to correspond to specific DNA sequences with recognizable PAMs.

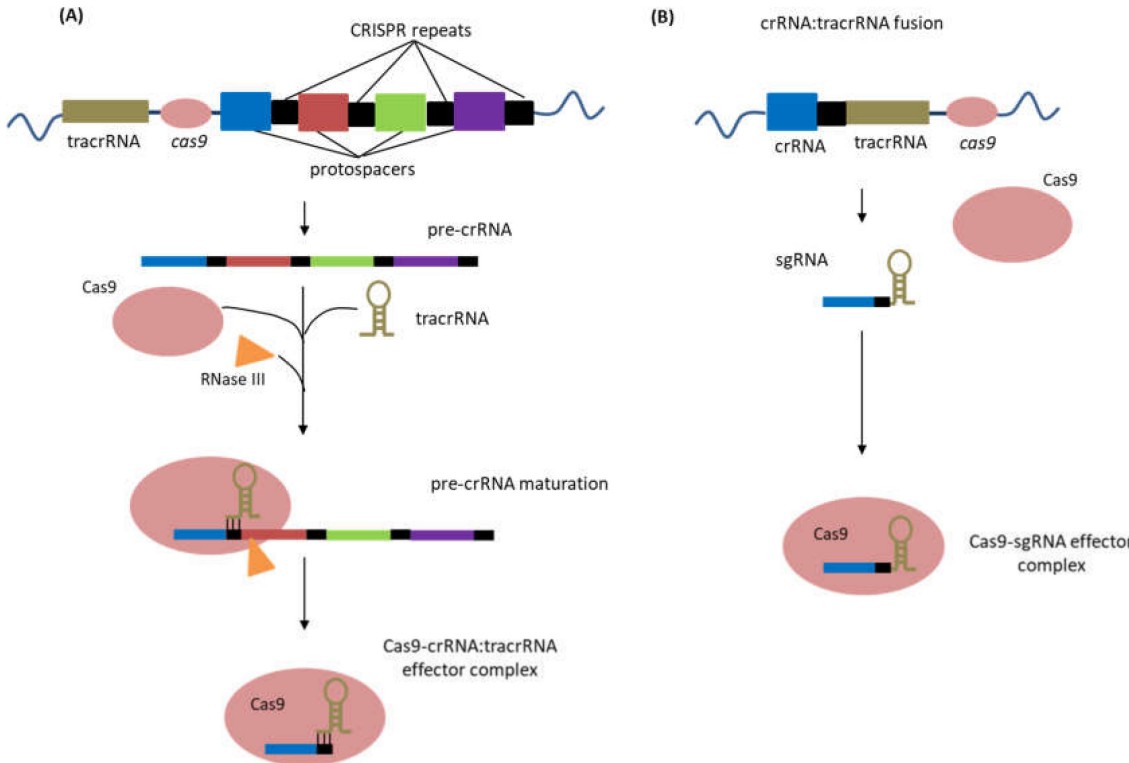

**Figure 3.** Simplified CRISPR-Cas9 system. The CRISPR-Cas9 system has been engineered to ease the implementation of technology as a genome editing tool in various organisms. (**A**) Naturally occurring CRISPR-Cas 9 system generates functional crRNAs from the transcription of protospacers in the CRISPR locus which hybridized with tracrRNA, facilitating the formation of mature crRNAs through the actions of the endogenous RNase III and Cas9. An effector complex comprising the crRNA:tracrRNA hybrid and Cas9 is then formed; this effector complex can recognize target DNA sequence upstream of a recognizable PAM and induce DSB. (**B**) The engineered CRISPR-Cas9 system used today consists of a construct of a sgRNA, which is a fusion of a crRNA and tracrRNA. The sgRNA forms a complex with Cas9 nuclease and together mediates the cleavage of the target DNA site.

## 3. Application of CRISPR-Cas9 in Plant Genome Editing

After the successful demonstration of CRISPR-Cas9 as a programmable RNA-guided editing system [36], a large number of studies have been conducted and proven the efficiency of this system in various organisms. The application of this platform has greatly influenced the advancement of genome manipulations in plant systems. Utilizing the ability of this genome editing platform to induce DSBs allows gene knockout and knockin in plants through the endogenous DSB repair pathways NHEJ and HR, respectively. NHEJ-mediated gene knockouts can be used to silence or eliminate genes that negatively affect plant growth and productivity. HR, on the other hand, can be exploited to integrate transgenes that could confer desirable traits in plants [37]. Early applications of CRISPR-Cas9-based genome editing in plants have demonstrated the versatility of the system in generating targeted gene mutations and corrections over a variety of plant species, ranging from model plants to some economically important crops [38–43]. CRISPR-Cas9 system has been rapidly developed as a tool for genome modifications and has been applied to improve plant quality and yield, alter metabolic pathways, confer stress tolerance, and to characterize gene functions (Table 1).

**Table 1.** Application of CRISPR-Cas9 in plant biology.

| Crop | Target Gene(s) | Outcome/Function | Reference |
|---|---|---|---|
| | Improving Crop Yield and Nutritional Content | | |
| *Oryza sativa* | *OsGS3, OsGW2* and *OsGn1a* | Improved yield (grain length, width, number and 1000-gain weight) | [44] |
| | *SBEIIb* | High amylose-containing rice | [45] |
| | *OsFAD2*-1 | Increase oleic acid content | [46] |
| | *TMS5* | Thermo-sensitive genic male sterile line | [47] |
| *Brassica napus* | *FAD2* | Increase oleic acid content | [48] |
| *Solanum lycopersicum* | *SGR1, LCY-E, Blc, LCY-B1 AND LCY-B2* | High lycopene content | [49] |
| | *SlyPDS, SlyGABA-TP1, SlyGABA-TP2, SlyGABA-TP3, SlyCAT9 and SlySSADH* | Accumulation of *y*-aminobutyric acid (GABA) content | [50] |
| *Zea mays* | *ZmTMS5* | Thermo-sensitive male-sterile lines | [51] |
| *Triticum aestivum* | *Ms1* | Thermo-sensitive male-sterile lines | [52] |
| | Generating Stress Resistant Crops | | |
| *Cucumis sativus* | *eIF4E* | Immunity to *Cucumber vein yellowing virus* (Ipomovirus) infection and resistance to potyviruses *Zucchini yellow mosaic virus* and *Papaya ring spot mosaic virus-W* | [53] |
| *Oryza sativa* | *eIF4G* | *Rice tungro spherical virus* (RTVS)-resistant rice | [54] |
| | *OsERF922* | Resistance against rice blast disease | [55] |
| | *OsRR22* | Salt tolerance | [56] |
| *Citrus sinensis* Osbek | *CsLOB1* | Canker disease resistant | [57] |
| *Citrus paradisi* Macf. | | | [58] |
| *Solanum lycopersicum* | *Mlo* | Resistance to powdery mildew | [59] |
| *Zea mays* | *ARGOS8* | Resistance to drought | [60] |
| | Functional Characterization of Genes | | |
| *Oryza sativa* | *MPK1* and *MPK6* | Essential genes for rice development | [61] |
| | *OsSWEET11* | Sucrose transporter | [62] |
| | *OsAnn$_3$* | Involved in cold tolerance | [63] |
| | *OsMADS3* | Regulator of flower meristem maintenance and determinacy | [64] |
| | GT-1 element in the promoter region of *OsRAV2* | Involved in salt induced expression | [65] |
| *Solanum lycopersicum* | *SIMAPK3* | Involved in drought tolerance | [66] |
| | *SICBF* | Involved in chilling tolerance | [67] |
| | *SIPHO1;1* | Phosphate acquisition and transfer | [68] |
| | *AP2a, NOR, FUL1* and *FUL2* | Fruit development and ripening | [69] |

### 3.1. Improving Crop Yield and Nutritional Content

Global food security is a prevailing issue that continues to be challenging due to the booming world population and the shift of dietary preferences. To meet these increasing demands, crop optimization regarding yield, nutritional content, and plant quality would be necessary. Although enormous efforts have been made to boost crop productivity through optimizing conventional agricultural technologies and employing high-end machinery, these options remain problematic due to their costly and time-consuming processes. Innovative approaches such as genome editing have been proposed as an alternative avenue to address current and potential agricultural challenges, in the hope of securing the food supply [70].

Numerous studies have demonstrated the promise of the CRISPR-Cas9 system in developing new desirable and heritable traits in various crops. Some of these studies include the generation of rice (*Oryza sativa*) elite varieties (J809, L237, and CNXJ) with improved yield (grain length, width, number, and 1000-gain weight) through simultaneous editing of three trait-related quantitative trait loci (QTLs) genes *OsGS3*, *OsGW2*, and *OsGn1a* [44]. High amylose-containing rice has been produced also via CRISPR-Cas9-mediated mutagenesis of starch branching enzyme *SBEIIb* [45]. Early-maturing cultivars of rice were developed as well through employing CRISPR-Cas9- mediated multiplex genome editing of three flowering suppressor genes *Hd2*, *Hd4*, and *Hd5* involved in the photoperiodic flowering pathway, allowing time- and labor-efficient breeding [45]. A significant increase in oleic acid content in rice seed [46] and oilseed rape (*Brassica napus*) [48] was also achieved through the successful mutation of fatty acid desaturase 2 (*FAD2*) genes, *OsFAD2*-1 and *FAD2*, respectively, through CRISPR-Cas9 system. Lycopene-enriched tomato (*Solanum lycopersicum*) has also been developed using this genome editing platform by altering the genes involved in the lycopene biosynthesis simultaneously [49]. In the same way, *y*-aminobutyric acid (GABA) content has been shown to accumulate in tomato by altering the genes involved in GABA shunt [50]. CRISPR-Cas9 system has been implemented in maize (*Zea mays*) [51] and wheat (*Triticum aestivum*) [52] as well to simplify hybrid maize and wheat seed production through silencing the thermo-sensitive genic male-sterile 5 (*ZmTMS5)* gene and wheat male fertility gene (*Ms1*), respectively, resulting in the generation of thermo-sensitive male-sterile lines. A similar approach has been reported to generate a thermo-sensitive genic male sterile line in rice [47]. Aside from demonstrating the successful implementation of CRISPR-Cas9 in improving crop quality, these studies also highlighted the stability of mutations that can be readily transmitted to the next generation, the specificity of the mutation, the generation of transgene-free crops, and the efficiency of the system to perform multiplex genome editing.

### 3.2. Biotic and Abiotic Stress Resistance/Tolerance

Plants are exposed to various climatic (abiotic) stressors and are susceptible to the infestation of a large set of pathogens (biotic) that adversely affect plant growth and development, leading to the deterioration of crop productivity [71]. Thus, enhancing the plant's resistance to these stressors plays a crucial role in regulating crop productivity to meet the demand of the growing global population [8]. To overcome these challenges, traditional disease control and conventional agricultural techniques have been utilized and found to be effective over decades. However, these conventional approaches have constraints and may not address the emerging challenges due to the evolving nature of pathogens and the current changing environmental conditions. Targeted genome editing through CRISPR-Cas9 has also been viewed in this respect as a promising approach to develop plant varieties that are resilient to biotic and abiotic stress factors [8,72].

Successful implementation of the CRISPR-Cas9 system in engineering plants to confer resistance or tolerance to various stresses has proven the concept to be effective and plausible. For instance, the generation of virus-resistant crop like cucumber (*Cucumis sativus*) through the disruption of eukaryotic translational initiation factor 4E (*eIF4E*) gene provided immunity to *Cucumber vein yellowing virus* (Ipomovirus) infection and resistance to potyviruses *Zucchini yellow mosaic virus* and *Papaya ring spot mosaic virus-W* [53]. Similarly, a *Rice tungro spherical virus* (RTVS)-resistant rice have been

developed using the system through targeting the translation initiation factor 4 gamma (*eIF4G*) gene [54]. Crops that are resistant to either bacteria or fungi pathogens have been developed as well like the canker-resistant Wanjincheng orange (*Citrus sinensis* Osbek) [57] and Duncan grapefruit (*Citrus paradisi* Macf.) [58], both harboring mutations in the promoter region of lateral organ boundaries 1(*CsLOB1*) gene—the susceptibility gene for citrus canker caused by *Xanthomonas citri*. The targeted knockout of the ERF transcription factor *OsERF922* gene in rice has resulted in augmented resistance against rice blast disease caused by the filamentous fungus *Magnaporthe oryza* [55]. A variety of tomato resistant to the powdery mildew fungal pathogen *Oidium neolycopersici* generated through CRISPR-Cas9-mediated mutagenesis of mildew resistant locus O (*Mlo*) that confers susceptibility to the fungus has also been reported [59].

Moreover, selected studies have shown the utilization of the CRISPR-Cas9 system in developing plant varieties that can tolerate adverse environmental conditions. Shi et al. [60] reported the generation of drought-resistant maize achieved through replacing the native promoter of the ethylene negative response regulator *ARGOS8* gene by a moderately constitutive maize promoter GOS2, via the HR repair pathway. This has led to an elevated level of *ARGOS8* expression, thereby increasing the yield under drought conditions. In a recent study, CRISPR-Cas9-targeted knockout of salt-related *OsRR22* gene that encodes a response regulator transcription factor involved in both cytokinin signal transduction and metabolism has been shown to enhance salinity tolerance in rice [56]. Indeed, the CRISPR-Cas9 genome editing system could potentially improve crop productivity by enhancing the crop's ability to resist and withstand both biotic and abiotic stress conditions. This has been achieved via targeted-alteration of disease susceptible genes or transcription factors that control the expression of genes involved in stress responses.

### 3.3. Functional Characterization of Genes and Regulatory Elements

CRISPR-Cas9 technology has been rapidly used to elucidate and study gene functions and regulatory elements through reverse genetic analysis approach. Multiple studies have documented the application of this technology in validating predicted functions of genes and gene homologs in model plants and some important agricultural crops. In rice, for instance, several genes have been functionally annotated using the CRISPR-Cas9 system. These include the mitogen-activated protein kinase (MPK)—related genes *MPK1* and *MPK6* as essential genes for rice development [61]; the essential role of *OsSWEET11* sugar transporter in releasing sucrose during the early stage of caryopsis development [62]; the involvement of rice annexin gene *OsAnn₃* in cold tolerance of rice [63]; and the discovery of *OsMADS3* gene as a floral organ number2 (*FON2*) gene co-regulator of flower meristem maintenance and determinacy in rice [64]. In addition, the CRISPR-Cas9 system has been used to validate in situ the regulatory function of the GT-1 element in the promoter region of the *OsRAV2* gene for salt-induced expression in rice [65]. In the study of abiotic stress tolerance in tomato, CRISPR-Cas9-targeted knockout of *SlMAPK3* gene, a member of *MAPK* gene family has resulted in reduced drought tolerance of the plant, revealing the essential involvement of *SlMAPK3* gene in drought tolerance [66]. Knockout of the *SlCBF* gene, a member of the C-repeat binding factors (CBFs) family, revealed its functional role in the chilling tolerance of tomato as the inactivation of the gene led to severe chilling injuries and pleiotropic effects on plant's stress responses [67]. A similar approach has led to the validation of a *SlPHO1;1* gene function, a gene homolog of phosphate 1 (*PHO1*) gene present in both vascular and non-vascular plants that plays an important function in phosphate acquisition and transfer [68]. Re-evaluation study of the transcription factors that control fruit development and ripening in tomato, which were previously characterized based on spontaneous mutants or RNAi knockdown lines, has been carried out also using CRISPR-Cas9 technology [69]. A study on maize using the CRISPR-Cas9 system to inactivate CCT transcription factor (*ZmCCT9*) highlighted the significance of the gene in maize adaptation to higher latitudes [73]. Similarly, an RRM_RBM48 type RNA binding protein encoded by *Dek42* in maize that affects pre-mRNA splicing during kernel development has been functionally confirmed by CRISPR-Cas9-mediated mutagenesis [74]. Hence, the reverse genetic

approach through CRISPR-Cas9 based mutagenesis paved the way for the understanding of biological processes and networks through rapid and straightforward inactivation of genes for in vivo gene function and regulatory components analyses.

## 4. CRISPR-Cas9 System: Hurdles, Existing Approaches, and Developments

The emergence of the CRISPR-Cas9 system as a tool for targeted genome editing has considerably changed the phase of plant biology research. Several studies have been circulating detailing the utility of this system in improving crops' quality and productivity through genome manipulation. The system also exerts great help in understanding how biological processes operate via rapid and stable gene mutagenesis for reverse genetic analyses. Although the system has been proven its versatility and functionality in plant genome editing, several considerations and limitations of the system are to be addressed and resolved for its effective implementation. The following sections summarize some important technicalities and parameters regarding the application of the technology and also highlight the current methodological advancements for the optimization of the system for improved implementation, on-target efficiency, and versatility.

### 4.1. sgRNAs-Cas9 Molecular Construct

Constructing the plasmid-based sgRNAs/Cas9 cassette is the critical point for the efficacy of the CRISR-Cas9 system as it determines the proper expression of the Cas9 nuclease and the specificity of the targeting sgRNAs. Several factors must be taken into account such as the choice of the target sequence, PAM compatibility, the promoter activity, as well as the overall architecture of the expression cassettes to efficiently employ the genome editing system in plants and to avoid possible off-target effects.

### 4.1.1. Target/gRNA Sequence

Directing the nuclease activity of Cas9 protein to a specific target DNA site hinges on the specificity of the gRNAs that are transcribed from short nucleotides complementary to a portion of the target DNA sequence. Proper identification of the target site and the appropriate design of the gRNAs are thus crucial for the accurate and efficient targeting of the CRISPR-Cas9 system. The construction of a single gRNA (sgRNA) typically requires 20 nt corresponding to the target DNA sequence followed by a recognizable PAM. Complementarity between sgRNA and target DNA bases dictates the affinity and efficacy of the Cas9 nuclease activity. Mismatches occurring at the region near the PAM can significantly reduce the Cas9 cleavage activity. Incompatibility of bases in this region can be tolerated up to two bases. However, the presence of mismatches at the 5′ end of the sgRNA-DNA paired region is better tolerated than at the end near the PAM [75,76]. It should also be noted that the target sequence contains no sequence similarity with other genes to prevent non-specific cuts in the genome. To date, many bioinformatics tools and web-based portals can be utilized to aid in designing highly specific sgRNAs in silico (e.g., COSMID, CRISPR-PLANT, GGGenome program, CRISPRdirect, CRISPR Genome Analysis Tool, and among others), allowing gene targeting with minimal to no off-target effects [77]. For comprehensive details of these tools, see the reviews of Cui et al. [78], Wilson et al. [79], and the recent work of Lui et al. [80].

It is noteworthy to know as well that the cleavage efficiency is highly dependent on the selected target sequence in the sgRNA; cleavage efficiency differs even between adjacent target sites [81]. In addition, a high or low GC content has also been shown to reduce the efficiency of cleavage [82]. As such, proper selection of target sequence in the target gene or employing multiple sgRNAs, each with different target sites on the gene of interest must be carried out to ensure efficient mutagenesis of the target DNA. An in vitro cleavage assay can also be performed to assess the cleavage efficiencies of the selected sgRNA sequences [83], to sort out effective and efficient sgRNAs before employing in vivo to maximize the chances of obtaining efficient mutation frequencies in the target gene as well as to avoid the risk of not generating mutant at the end of a time-consuming process.

### 4.1.2. PAM Compatibility

The PAM sequence is a 3-5 nucleotide motif that serves as a recognition signal for Cas9 and is strictly required for Cas9-mediated cleavage of target DNA. The availability of PAM sequence location in a gene determines the choice of sgRNA sequence and thus limits the range of the DNA sequence that can be targeted by the CRISPR-Cas9 system. Thus far, the *S. pyogenes* Cas9 (SpCas9) is the commonly used Cas9 that recognizes NGG PAMs and is thereby restricted to target sites that contain this motif [36]. This PAM requirement of SpCas9 is of particular concern, especially when editing gene through HDR that requires precise location of the DSBs that are placed in close proximity (10–20 bp) to the region of interest or when doing nucleotide specific mutagenesis [84]. To circumvent this constraint, several approaches have been employed such as the use of SpCas9 that has been purposely engineered to recognize alternative PAM sequences and the use of Cas9 nucleases derived from bacteria other than *S. pyogenes*.

#### SpCas9 Variants

It has been demonstrated that mutation of the PAM-interacting domains of wildtype SpCas9 could lead to SpCas9 derivatives with varied PAM specificities. Such a mutation strategy has led to the development of SpCas9 variants capable of recognizing NGAN, NGNG, NGAG, and NGCG PAM sequences [85]. These variants have been shown to efficiently modify genomic loci that cannot be modified using the wildtype SpCas9 and can be incorporated into existing SpCas9 vectors through site-directed mutagenesis. xCas9, another engineered SpCas9 variant has also been reported to expand the PAM compatibilities of SpCas9 which recognizes a broad range of PAM sequences including NG, GAA, and GAT with substantially high DNA specificity and lower off-target activity compared to its native counterpart [86]. Successful implementation of SpCas9 variant xCas9 and its derivatives, and other variants such as SpCas9-NG, and SpCas9NGv1 has been reported to efficiently target endogenous genomic sites in plants such as rice and *Arabidopsis* [87–92]. Moreover, it has been shown that xCas9 is a high-fidelity nuclease, exhibiting high targeting activity at sites with canonical NGG PAMs, while SpCas9-NG variant exhibited robust editing activity at sites with relaxed NG PAMs without any preference for the nucleotide following NG [87,89] same as true with SpCas9NGv1 variant [88].

SpRY, another SpCas9 variant recently reported by Walton et al. [93], has been shown to recognize and edit genomic loci bearing nearly all PAMs. This near-PAMless variant exhibited robust activities on a wide range of target sites with NRN PAMs (where R is A or G). Substantial activities on sites with NYN PAMs (where Y is C or T) were also observed albeit lower compared to sites with NRN PAMs. However, the expanded PAM recognition of this variant has led to an increased tendency for off-target editing. Engineering the SpRY into a high-fidelity SpRY (SpRY-HF1) has eliminated nearly all off-target editing events and substantially improved the on-target efficiency. It can be noted that SpRY-HF1, with its improved fidelity, can be employed for genome editing applications that necessitate high specificity. Although the targeting capabilities of SpRY have been demonstrated only in human cells, the development of this near-PAMless variant indeed further offers a new genome editing capabilities and improved editing resolution that can be applied in a variety of biological systems, including plants in the near future.

Recently, the use of an artificial inhibitory domain (ARC) fused to Cas9 has been shown to fine-tune the target specificity of the editing system in mammalian cells [94]. This kinetic control of the Cas9 editing reagents may be incorporated into the existing SpCas9 variants to further enhance the targeting specificity, thus reducing the likelihood of generating unwanted genome editing events.

#### Cas9 Orthologues

Several Cas9 orthologues from other bacteria have been exploited as well to expand the targeting scope of the CRISPR-Cas9 system for plant genome editing. Cas9 orthologues from *Staphylococcus aureus*

(SaCas9) and *Streptococcus thermophilus* (St1Cas9) have been reported to efficiently modify genes in *A. thaliana* via NHEJ targeted mutagenesis with mutation frequencies comparable to SpCas9, and resulted in a stable and heritable mutation [95,96]. On the other hand, a comparative study on the efficiency of different Cas nucleases for *Nicotiana benthamiana* gene editing has demonstrated that SaCas9 is highly efficient among the rest of the Cas nucleases tested including SpCas9 [97]. Both StCas9 and SaCas9 orthologues are smaller in size compared to SpCas9, which makes them suitable alternatives especially when small vectors are used such as viral expression vectors [98]. Moreover, these orthologous systems have different requirements concerning the interaction between crRNA and tracrRNA, the target sequence, and the PAM compatibility. StCas9 recognizes the PAM sequences NNAGAA and NNGGAA with similar induction frequencies of mutation for both PAMs. SaCas9 on the other hand can recognize NNGGGT and NNGAA PAM sequences with greater mutation yields on the former PAM sequence, suggesting that the mutagenesis outcomes may differ on the selected PAMs. As these orthologues require longer PAM sequences in contrast to SpCas9, they should be expected to significantly reduce the frequency of off-target effects compared to SpCas9 [95].

These studies benchmarked the SpCas9 variants and SpCas9 orthologues as versatile genome engineering tools that significantly enhanced and broadened the PAM compatibility of the CRISPR-Cas9 system for plant genome editing. Researchers attempting to perform genome editing in plants using the CRISPR-Cas9 system can readily choose from these Cas9 variants or orthologues according to their specific PAM compatibility needs.

### 4.1.3. Choice of Promoter and Expression Cassette Structure

The expression levels of CRISPR-Cas9 components (sgRNAs and *cas9*) have been shown to be highly associated with the efficiency of the system. In addition, the architecture of the individual expression cassettes also holds a significant impact on the efficiency of the system (Figure 4). Thus, the choice of the promoter to drive the expression of *cas9* and sgRNAs, and how these expression cassettes are configured in the overall expression system are important to consider for efficient genome editing.

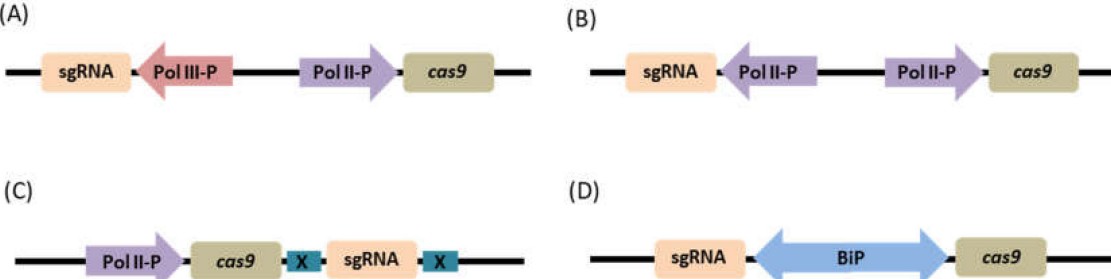

**Figure 4.** CRISPR-Cas9 expression cassette structures. (**A**) Mixed-dual promoter system. The typical expression cassette structure of CRISPR-Cas9, consisting of two different classes of promoters, Pol II and Pol III promoters controlling the expression of *cas9* and sgRNA, respectively. (**B**) Dual Pol II system. The *cas9* and sgRNAs are expressed in separate expression units under the control of two independent Pol II promoters. (**C**) Single transcriptional unit. Both the *cas9* and sgRNA are expressed as a single transcript under the control of a single Pol II promoter. The RNA processing system (X: csy4, ribozyme, tRNA) is usually incorporated in the construct to separate sgRNA from *cas9* after transcription. (**D**) Bidirectional promoter system. A bidirectional promoter (BiP) which can initiate transcription in both orientations is used to express both *cas9* and sgRNA.

### Mixed Dual Promoter System

Transgene expressions in plants are most frequently achieved through the use of strong constitutive RNA polymerase II—based (Pol II) promoters such as the cauliflower mosaic virus 35S (*CaMV35S*) promoter, the rice *Actin1* and cytochrome *c* gene promoters, and the maize ubiquitin 1 gene (*Ubi1*) promoter [99–101]. For the CRISPR-Cas9 system, these promoters have been used to initiate the

expression of the *cas9* gene and reported consequently to work efficiently, however the frequency of mutation varies depending on the plant species [43,102,103]. Another class of promoters, the RNA III polymerase-dependent (Pol III) *U6* promoters that drive high expression levels of small nuclear RNAs (snRNAs) in plants [104], has been frequently chosen to drive sgRNA expression in CRISPR-Cas9 vectors [39]. This conventional configuration wherein the two different expression cassettes for *cas9* and sgRNA are driven by two independent promoters, usually of different classes is referred to as the mixed dual promoter system.

The efficiency of *U6* promoters for sgRNA expression has been successfully demonstrated in various plant species with the *U6* rice promoters *OsU6a*, *OsU6b*, and *OsU6c* being the most commonly used for monocots, and for dicots the *Arabidopsis AtU6*-1 and *AtU6*-29 promoters are the most preferred ones [43,105,106]. Despite the functionality of the exogenous *U6* promoters, it has been apparent that the utilization of endogenous or species-specific *U6* promoters can result in an increased sgRNA expression and thereby can improve the editing efficiency. For instance, the use of an endogenous *GhU6* promoter for CRISPR-Cas9-mediated mutagenesis in cotton has resulted in an increased sgRNA expression levels up to 6-7 times higher than the *Arabidopsis AtU6*-29-driven sgRNA, and also improved the mutation efficiency 4-6 times [107]. Similarly, in soybean, the endogenous *GmU6* promoter has doubled the sgRNA expression levels compared to those obtained using the *Arabidopsis AtU6-26* promoter, and thus led to an enhanced gene editing efficiency (14.7–20.2%) [108]. Plant genome houses multiple *U6* genes of which their promoters could potentially be exploited, however, their expression levels vary to some degree and thus imply that not all endogenous *U6* promoters are equally efficient in driving gene expression [109]. Both exogenous and endogenous *U6* promoters, together with the commonly used constitutive promoters must be sufficiently characterized to maximize CRISPR-Cas9-mediated mutation efficiency in specific plant species.

Several promising Pol II promoters for enhancing the efficiency of the CRISPR-Cas9 system have been reported recently, such as the use of tissue-specific promoters like the cell division-specific promoters to control the expression of the *cas9* gene. The study of Feng et al. [110] demonstrated the efficacy of *pCDC45* promoter producing 80.9% to 100% mutation frequencies in *Arabidopsis* T1 generation. In addition, the promoter also achieved higher mutation efficiencies (60.17%) when used to simultaneously target multiple genes (multiplex CRISPR-Cas9 system) compared to the constitutive ubiquitin promoter (43.71%); higher efficiency of heritable mutations was also observed. In maize, the meiosis-specific promoter of the *dmc1* gene has been shown to generate highly efficient homozygous and bi-allelic mutations (66%) in T0 plants which can also be stably transmitted to the next generations. The efficiency of the *dmc1* promoter has been attributed to its high activity, resulting in a concurrent high *cas9* expression. Surprisingly, the activity of the *dmc1* promoter, which was thought to be specifically expressed in the reproductive tissues, was also highly active in other maize tissues including the callus. In the same study, they demonstrated that sgRNAs driven by endogenous *U3* promoters combined with the *dmc1*-controlled *cas9* can achieve highly efficient mutations compared to *U6*-driven sgRNAs. The evolutionary conservation of the *dmc1* gene suggests that exploiting the *dmc1* promoter for the CRISPR-Cas9 system could be functional and effective in other plant species [111].

Dual Pol II Promoter System

Although Pol III promoters *U3* and *U6* are convenient to use for sgRNA expression, these promoters are not well characterized, especially in non-model plants. Additionally, these promoters are used to express short sequences by nature, usually perform poorly when used in heterologous expression, and could not match the expression strength of some stringent Pol II promoters [108]. It would be desirable to have both *cas9* and sgRNA under the control of the same class of promoters, like Pol II to achieve a coordinated expression. Pol II promoters for sgRNA expression can provide flexible options in terms of spatiotemporal control of expression in vivo when needed. In addition, Pol II promoters can produce longer transcripts since the activity of Pol II is not hindered by the presence of short

internal termination sites [112], this feature of Pol II promoters is important when expressing multiple stacked sgRNAs.

In the dual Pol II promoter system, the *cas9* and sgRNAs are expressed in separate expression units under the control of two independent Pol II promoters. This strategy has been demonstrated by Cermak et al. 2017 [113]. The construct was composed of a *cas9* expressed under the control of *CaMV35S*, and to avoid duplicate use of promoters, the Pol II Cestrum Yellow Leaf Virus promoter (*CmYLCV*) was used to drive the expression of sgRNA. The *CmYLCV* promoter is a strong constitutive promoter and has been known to have comparable strength to that of the *CaMV35S* or *Ubi1* promoters when used in a variety of crops, both dicots and monocots [114]. High genome editing efficiency was obtained using the Pol II-driven sgRNA system in tomato compared to the conventional sgRNA expressed through Pol III promoters. The overall mutagenesis frequency was found to be twice higher than the construct with Pol III promoters. The system also exhibited the same high editing efficiency when used to modify genes in *Medicago truncatula.* The same strategy was also applied in wheat and barley in which the *cas9* was driven by Pol II *Ubi1* promoter and the sgRNA was expressed through *Panicum virgatum* ubiquitin Pol II promoter *PvUbi1*, the system also achieved high editing frequencies of the target genes [113]. The high editing efficiency has been credited to the increased expression of sgRNAs when controlled by a strong Pol II promoter.

One important consideration in using Pol II promoters for sgRNA expression is the incorporation of appropriate sgRNA processing systems in the molecular construct. Generally, transcripts produced by Pol II promoters are modified by 5′ capping and 3′ poly-A tail addition and subsequently transported to the cytosol from the nucleus. These post-transcriptional modifications and altered localization could potentially affect the efficient use of sgRNAs. Hence, the incorporation of sgRNA processing systems would allow the excision and processing of sgRNAs from the Pol II transcripts into mature sgRNAs [115]. Details on these sgRNA processing systems are discussed in the succeeding texts, and the context of multiplexed genome editing.

Single Transcriptional Unit

In most studies regarding CRISPR-Cas9 for plant genome editing, the utilization of a mixed dual promoter system is common in which the *cas9* and sgRNAs are expressed separately by a committed Pol II and Pol III promoters, respectively. The system works pretty well for gene editing wherein a lone sgRNA has to be expressed by one promoter. However, for multiplexed genome editing where multiple sgRNA expression units are stacked, the use of this mixed dual system is challenging. Since individual sgRNA has to be with its own promoter, the construct could add up to the length of the expression vector which in turn complicates the delivery of the whole construct into the host plant and may trigger gene silencing [116,117]. Moreover, the use of multiple promoters like U6 and U3 promoters for multiple sgRNAs in one construct has been shown to cause variations on the expression levels of the sgRNAs, resulting in noticeable variations on the editing efficiency [118]. The development of a stacked sgRNA expression system comprising multiple sgRNAs expressed from a single promoter has somehow circumvented the existing limitations and has improved the system. In this system, the array of sgRNAs in the molecular construct is spaced by sequences that are recognized by RNA-cleaving enzymes which process post-transcriptionally the polycistronic sgRNA transcripts into individual sgRNAs. Thus far, the commonly used spacer sequences are those that are recognized by the host endogenous tRNA processing enzymes [113,119], the CRISPR-associated RNA endoribonuclease Csy4 [113,120], and ribozymes [121]. However, these constructs are still composed of *cas9* and sgRNAs that are expressed in two separate expression units driven by two different promoters Pol II and Pol III for *cas9* and sgRNAs, respectively. Coordinated expression of *cas9* and sgRNAs is thus difficult to achieve. A more advanced configuration has been established comprising a single expression unit of a *cas9* and sgRNAs expressed through a single promoter, preferably Pol II promoter. Tang et al. 2016 [122] demonstrated the use of a single transcript unit (STU) CRISPR-Cas9 system for both single-gene target and multiplexed editing in rice, *Arabidopsis*, and tobacco with high

efficiency of mutation. The co-expression of *cas9* and sgRNAs is driven by Pol II promoters and the separation of *cas9* transcript with sgRNAs as well as the processing of sgRNAs is based on hammerhead (HH) ribozyme. They tested several known stringent Pol II promoters such as *CaMV35S* and *Ubi1*, and have shown efficient mutagenesis in varying degrees, which emphasized the direct correlation between mutation efficiency and the choice of Pol II promoter. The STU system has been shown to be more superior in genome editing compared to the conventional mixed dual promoter system [123]. More detailed information on STU in the context of multiplexed genome editing is presented in the later section of the review.

Bidirectional Promoter System

Optimal production of the editing components *cas9* and sgRNAs are always viewed as the key factor for successful and efficient genome editing. While the STU system offers a simple and compact expression system, relying on a single Pol II promoter to drive *cas9* and gRNA(s) expression, refinement of the system might be a hurdle. Given that the *cas9* and sgRNA are co-expressed in one transcript, the expression system may not be optimal for the production of the functional editing complex. Post-transcriptional processes might affect the translation of Cas9 protein and/or the proper processing of the sgRNA from the same transcript. Hence, obtaining an equal molar ratio of Cas9 and sgRNA might be difficult to achieve [124]. A bidirectional promoter (BiP), which can initiate transcription in both orientations, has been recently adopted for the CRISPR-Cas9 expression system. Similar to the STU system, only one promoter is required to express both *cas9* and sgRNA. However, the two components are positioned in a manner that they are transcribed in the opposite direction, one on each end of the promoter. The divergent architecture of the construct could balance the expression strength and would allow independent fine-tuning of the individual *cas9* and sgRNA expression cassettes with the use of different enhancers, 3′ UTR and terminators. Ren et al. 2019 [124] demonstrated the feasibility of the construct in expressing the editing reagents in transgenic rice. Initially, they utilized an engineered BiP system based on a double *CaMV35S* minimal promoter and an *Arabidopsis* enhancer. The two *CaMV35S* promoters flanking an *Arabidopsis* enhancer sequence were positioned in opposite directions, one driving *cas9* and the other one coordinating sgRNA expression. Csy4 was incorporated for the precise processing of sgRNA. The use of this system obtained detectable mutations with 20.7% and 52.9% editing efficiencies at two target sites in stable transgenic rice. The group further improved the system by using an endogenous BiP *OsBiP1* [124], a constitutive BiP of high expression in rice that drives expression of *Os02g42314* at the 5′ end and *Os02g42320* at the 3′ end. The endogenous BiP expression system with either tRNA and Csy4 sgRNA processing system has resulted in 75.9–93.3% editing efficiencies. The increased gene editing efficiency has been attributed to the observed expression strength driven by the endogenous BiP compared to the engineered double-mini *CaMV35S* promoter, suggesting that endogenous BiP is more efficient than the engineered BiP. However, this could not be always the case, the fact that both BiP systems can be further modified for improved efficiency. Ren et al. 2019 [124] also elaborated some strategies that can potentially improve the BiP system given its editing efficiency: (1) rational design-based approach to engineer BiP systems with improved functional strength, (2) identification and testing of endogenous BiPs, and (3) fine-tuning of *cas9* and sgRNAs using different combinations of 3′-UTR and terminator sequences. Future exploration of the BiP systems is expected to stem from these existing fronts.

Constitutive Versus Inducible

The high tandem expression of *cas9* and sgRNAs, in general, determines the targeting efficiency of the system; however, this could also lead to a high occurrence of off-target mutations. Constitutive expression of *cas9* and sgRNAs could lead to the persistence of the editing system, providing a wider opportunity to modify off-target sites of the genome, resulting in an increased incidence of non-specific mutations as demonstrated by studies in maize [103] and rice [125] plants using constitutive *cas9*. Although the targeting of non-specific sites are usually occurring at lower

frequencies than the defined target sites, the constitutive or tissue-specific expression systems could be more permissive to non-specific targeting by providing strong and continuous doses of Cas9 or sgRNAs for a period longer than the necessary. Nandy et al. 2019 [126] recently described an inducible expression system for controlling CRISPR-Cas9 mutagenesis in rice. The system used a heat-shock-inducible promoter to express *cas9* in tandem with the rice *U3* promoter for sgRNAs. Efficient mutations of the target gene were observed upon heat-shock treatment, obtaining ≥50% mutation frequencies in comparison with strong constitutive expression system. Notably, the system rarely induced mutations at non-permissive temperatures and targeted mutations were stably transmitted to the progeny at high frequency. Analysis of off-target mutations revealed undetectable or lower rate of off-targeting in the inducible expression system compared to the constitutive one. The inducible expression system can potentially curb the off-target effects and can offer spatiotemporal control of the editing reagents thereby avoiding possible toxicity and lethality during the developmental stages. Overall, the inducible system could significantly increase the precision and efficiency of mutagenesis while avoiding or keeping the off-target effects minimal through temporal control of expression and conditional targeting.

## 5. Multiplexing Strategies

One of the important features that put an edge on the CRISPR-Cas9 system is its programmable ability to simultaneously target multiple loci—the so-called multiplexed genome editing. To enable multiplexed genome editing using this system, multiple sgRNAs are required to be expressed simultaneously along with the Cas9 protein. Several different approaches have been developed and are in current use for simultaneous and efficient production of multiple sgRNAs; facilitating a straightforward multiplexed genome editing (Figure 5).

### 5.1. Csy4, Ribozyme, and tRNA-Based Configuration

Initial implementation of the multiplexing strategy was achieved by delivering multiple sgRNAs and Cas9 encoded independently in different multiple plasmids [41]. This strategy seemed to work, however, this eventually became troublesome and resulted in poor efficiency of editing due to the associated toxicity and burden upon the introduction of large amounts of plasmids to the host cells [127]. A later version of multiplexed genome editing using the CRISPR-Cas9 system was composed of expression vectors with an array of sgRNAs. In this construct, multiple sgRNAs with its dedicated promoter (usually Pol III) and terminator are joined in one expression vector alongside with *cas9* expression cassette [50,116,128]. While the construct is more straightforward than introducing multiple plasmids, the size of the final vector is a concern as the multiple sgRNA cassettes could lengthen the overall construct and would be difficult to deliver. Promoter cross talk may also potentially occur as multiple Pol III promoters are present, and in the worst-case gene silencing is likely to occur as well [127]. An alternative to this approach is the use of an expression unit that compacts multiple sgRNAs into a single transcript (polycistronic), which will then be processed into individual sgRNA by a dedicated processing enzyme, such as Csy4 endoribonucleoprotein, ribozymes, and the endogenous tRNA-processing RNases [113,120,121]. Instead of having multiple promoter-terminator formats to drive and delineate respectively the expression of individual sgRNAs, spacer sequences are introduced in between sgRNAs, and the whole sgRNA expression system is driven by a single promoter. These spacer sequences are recognized by its respective processing enzymes, allowing excision of the individual mature sgRNAs from the polycistronic transcript.

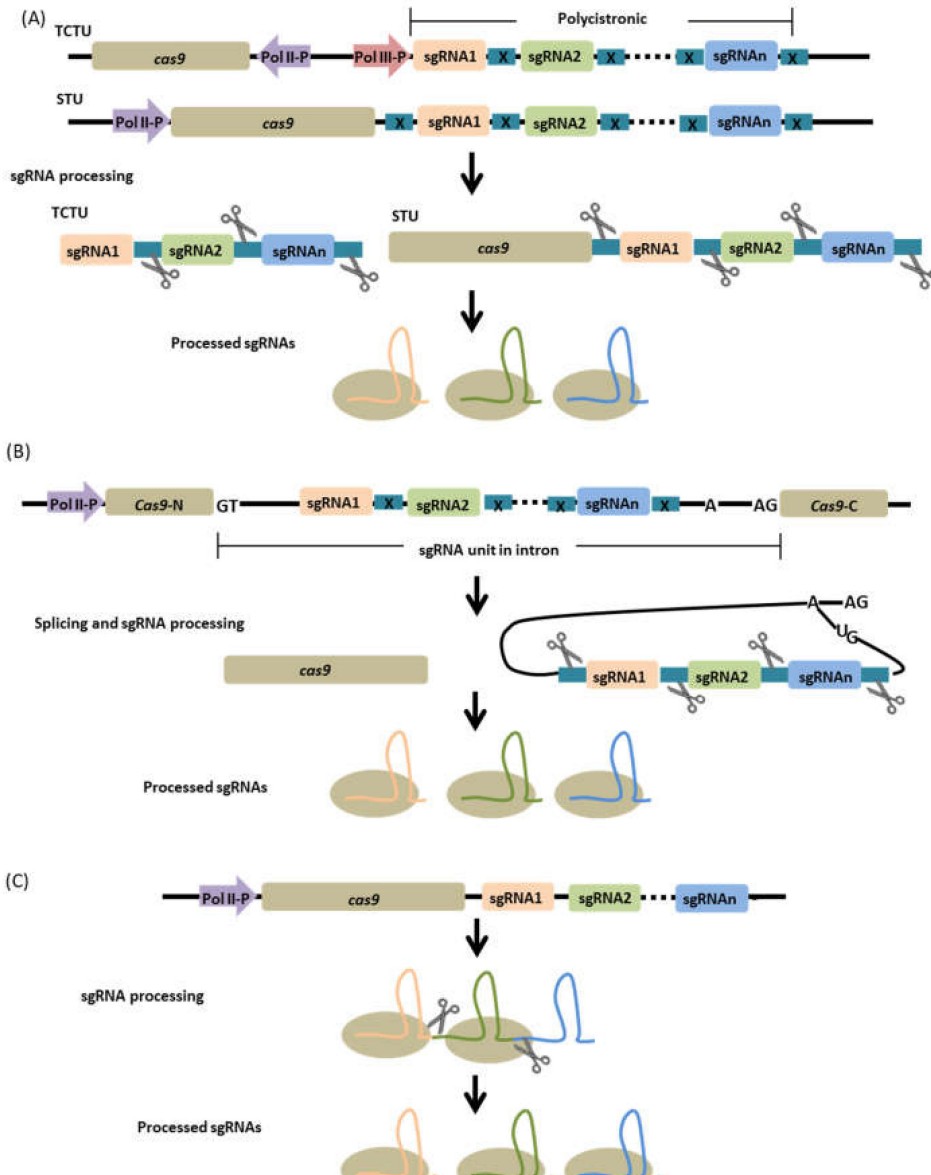

**Figure 5.** Current CRISPR-Cas9 multiplexing strategies. (**A**) Csy4, ribozyme, and tRNA-based configuration. The CRISPR-Cas9 vector is composed of an expression unit with multiple sgRNAs compacted into a single transcript (polycistronic). Spacer sequences (X) are introduced in between sgRNAs and the whole sgRNA expression system is driven by a single promoter. These spacer sequences are recognized by its respective processing enzymes (csy4, ribozyme, tRNA RNases), allowing precise excision of the individual mature sgRNAs from the polycistronic transcript. This design can be of two types: two-component transcriptional unit (TCTU) and single transcriptional unit (STU) largely based on the architecture of *cas9* and sgRNAs expression units in the vector. (**B**) Intron-based configuration. Introns are also engineered to carry stacked sgRNAs with dedicated RNA processing systems (X). The intron can be inserted into the ORF of the *cas9* gene and is transcribed along with the *cas9*. The intron is then spliced from the primary transcript forming the typical lariat structure. Individual sgRNAs will be processed by the RNA processing system and form complexes with Cas9. (**C**) Null-processing sgRNAs. A simplified vector format for multiplex genome editing that consists of *cas9* and sgRNAs in a single transcript without any incorporated sgRNA processing machinery. The array of sgRNAs is connected to the 3′ end of the *cas9* gene separated by 6-12 bp linkers. Accordingly, the translated Cas9 protein can be bound to the sgRNA transcript and the RNA sequences are trimmed by endogenous RNases consequently forming individual sgRNAs complexed with Cas9.

The Csy4 endoribonuclease was initially identified and characterized in the bacterium *Pseudomonas aeruginosa* as a constituent of the bacterium's type I-F CRISPR-Cas system. Csy4 is responsible for the generation of mature crRNAs through enzymatic cleavage at the 3′ bottom side of the stem-loops of the pre-crRNAs [129]. The specificity and activity of the cleavage mediated by Csy4 are dictated by the presence of 16-nt minimal RNA fragment, consisting of a stem-loop structure and one nucleotide downstream of the stem-loop. This 16-nt fragment is present in the 28-nt repeat sequence encoded by the CRISPR locus [129,130]. Given that the 16nt recognition site is sufficient for Csy4-mediated cleavage, exploiting the processing ability of Csy4 is possible for CRISPR-Cas9-based multiplexed genome editing by simply flanking the sgRNAs with the Csy4 recognition site and the co-expression of the Csy4 protein. The incorporation of the Csy4 cleavage site would allow proper excision and processing of stacked sgRNAs through the catalytic activity of Csy4 ribonucleoprotein. This multiplexing strategy was first successfully demonstrated in human cells [120], and was later adopted in plants to perform multiple-targeted mutagenesis in *Arabidopsis* [131], tomato, *M. truncatula*, wheat, and barley [113]. Ribozymes on the other hand are RNA molecules with enzymatic activities. Some are known to possess nuclease activities that facilitate the cleavage of RNA molecules at specific sites [132,133]. Taking advantage of their nuclease activity would thus permit the production of multiple sgRNAs from a polycistronic transcript with recognizable ribozyme-cleavage sites. Similar to the Csy4-based format, the concept was to flank the sgRNA with ribozymes. With the intrinsic nuclease activities of the ribozymes, the ribozyme-flaked sgRNA will undergo self-catalyzed cleavage, releasing mature sgRNA to complex with Cas9 [134]. To catalyze cleavage, all known ribozymes require a conserved sequence or need secondary structures downstream or around the cleavage site, respectively. Of known ribozymes, the hammerhead and hepatitis delta virus (HDV) ribozymes are commonly used for the construction of the ribozyme-sgRNA-ribozyme (RGR) unit [121]. The small hammerhead ribozymes, with their less than 50 bp in length core sequence corresponding to the stem-loop structures, are usually used at the 5′ end of the sgRNA in the RGR unit. For the 3′ end ribozyme, the HDV ribozymes are the most preferred choice because they require no specific sequence upstream of the cleavage site, allowing no alteration in the sgRNA scaffold [134]. This ribozyme-based strategy has been shown to successfully edit genes in plants [135,136] and has been the bases for multiplexing technology. Another approach for the production of multiple sgRNAs is to harness the endogenous tRNA processing system of the host cell. The RNA-processing systems are conserved in different organisms and are essential in the precise production of RNA—a fundamental cellular component. Engineering an artificial transcript that would allow recruitment of the endogenous RNA-processing system can thus facilitate the simultaneous production of multiple sgRNAs through precise cleavage of the primary transcript into individual sgRNAs [119]. To date, the well-characterized tRNA processing system has been exploited in this context. Transfer RNA (tRNA) is a cloverleaf-structured RNA molecule that facilitates the synthesis of protein in the ribosomes. The processing of tRNA from its precursor pre-tRNA involves the removal of extra 5′ and 3′ sequences through the action of RNase P and RNase Z, respectively [137]. To hijack the processing activity of the RNase P and Z for multiplexing strategy, the entire tRNA sequence is required and has to be incorporated in the polycistronic gene. The resulting polycistronic construct is composed of tandem repeats of tRNA-sgRNA with the tRNA sequence comprising the 6bp 5′ leader (5′-AACAAA-3′) and the 71bp mature tRNA [113,119,138]. The endogenous tRNA-processing RNases would recognize and cleave the tRNA components, allowing excision of individual sgRNAs from the polycistronic transcript. Aside from the robust processing of sgRNAs, the presence of tRNA in the molecular construct has been shown to enhance the efficiency of mutagenesis compared to the typical sgRNA construct [119]. tRNA serves as a transcriptional enhancer for Pol III owing to the presence of the internal promoter elements box A and B in the tRNA gene, allowing recruitment of the RNA Pol III complex [139]. Although all of these approaches have been proven to process multiple sgRNAs, the Csy4 and tRNA-based configuration have been shown to be superior compared to the ribozyme-based format [113,123].

Up to date several versions of these multiplexing strategies have been developed and can be categorized into two types: (1) two-component transcriptional unit (TCTU) and (2) single transcriptional unit (STU), categorized largely based on the architecture of *cas9* and sgRNAs transcriptional units (Figure 5). As with the mixed-dual promoter system, the TCTU system is composed of two different expression cassettes one for *cas9* and another for the polycistronic sgRNAs with incorporated processing systems. The promoters of each cassette could be either of the same type or different classes. Most of the early developed CRISPR-Cas9 multiplexing strategies that are based on polycistronic sgRNAs are of this category and have proven to efficiently target multiple genes in various plants [113,119,131]. Like the pioneering polycistronic TCTU system, the same known processing enzymes are incorporated in the STU system to excise and process multiple sgRNAs. However, in the STU system the *cas9* and the sgRNA cassettes are expressed through a single promoter, resulting in the generation of a single transcript comprising the *cas9* mRNA and the multiple sgRNAs. This configuration has been developed to achieve coordinated expression of *cas9* and sgRNAs, which is difficult to achieve using the TCTU. The system also offers a simpler configuration of the expression vector, without the need for laborious characterization of the promoters to be used as one strong Pol II promoter could drive the whole expression system. Several studies have demonstrated the superior ability of the STU system to perform multiplexed genome editing in plants, even compared with the TCTU system [122,123].

### 5.2. Intron-Based Configuration

Precise processing of sgRNAs is a crucial factor for the success of the multiplexed genome editing strategy. Engineering and exploiting different processing systems have greatly improved the processing and production of numerous sgRNAs from a single polycistronic transcript. In most of the above-mentioned strategies, especially the STU systems, the sgRNA cassette is linked at the 3′ end of the *cas9* gene with poly A or linker sequence in between. This architecture assures that the *cas9* is not immediately terminated by a terminator, allowing the continuous transcription of the sgRNA component. Moreover, the addition of 3′ poly A linker sequence allows proper translation of *cas9* transcript by providing mRNA stability and facilitating nuclear export. Although the efficacy of this format has been tested, possible negative effects on the *cas9* mRNA maturation and sgRNA stability may occur. An alternative to this design, introns can be customized to express sgRNAs that can be positioned at the 5′ end of the *cas9* or within its open reading frame (ORF). This intron-based design represents a more typical gene structure in eukaryotes for the expression of both editing components *cas9* and sgRNAs in a single mRNA [140,141].

Introns are non-coding regions of a gene that are spliced out from the primary mRNA prior to translation [142]. Present in some introns of coding genes are noncoding RNAs like small nucleolar RNAs (snoRNAs) and microRNAs. These non-coding RNAs are known to participate in some biological activities. Introns are turned into lariat RNA structures when spliced from primary mRNA; hence the presence of processed intronic noncoding RNAs implicates a dedicated processing pathway that uses introns as precursors [142]. Based on this mechanism, expressing sgRNAs from introns is possible when coupled with dedicated processing machinery to precisely excise individual sgRNAs from the host intron without altering normal splicing. This concept was first given light by Ding et al. 2018 [141] using the rice gene *Ubiquitin10* intron and tRNA processing system for sgRNA processing. In their molecular construct, the sgRNA-tRNA fragment was placed in between the 5′ splice site and branch site (5′-*GU-A-AG*-3′) of the intronic region of the *Ubiquitin10* gene. The sgRNA-containing intron was then positioned 5′ of the cas9 gene and the whole cassette was driven by the *Ubiquitin10* native promoter. With this configuration, the sgRNA-containing intron will be spliced from the primary transcript, resulting in the formation of the mature mRNA of Cas9 for protein translation, and the sgRNAs will be processed by the tRNA processing system from the spliced intron precursor subsequently enabling genome editing. Indeed, proper splicing and translation were observed, suggesting the compatibility of the construct with mRNA splicing. Efficient genome editing of the target genes in rice was also observed and the efficiency can be further enhanced by combining the *cas9* gene and sgRNA-containing

intron in different ways. Recently, a modification of this strategy has been presented by Zhong et al. 2020 [140]. In this study, introns embedded in gene bodies were used to express sgRNAs and inserted into the ORF of the *cas9* gene, breaking apart the gene's ORF. Three introns from different plant species were used: 189-bp *StIV2* intron (inS) from potato, a 290-bp *OsCDPK2*_1 intron (inO) from rice and a 190-bp *RcCAT*_1 intron (inR) from castor bean. These introns were modified to carry sgRNA coupled with tRNA, ribozyme, or with no processing system and inserted into the ORF of SpCas9. Targeted genome editing was achieved using the constructs, suggesting precise splicing and correct joining of exons. Among the sgRNA processing system used, the tRNA-based system has been shown to be robust, resulting in a high editing frequency. The system has also been shown to efficiently perform when used for multiplexed genome editing, however decreased in efficiency has been observed as more sgRNAs are inserted into the intron. This effect was speculated as a result of using small intron size. It has then proposed to alternatively use multiple introns to carry sgRNAs for a more effective multiplexing strategy. Additionally, the use of large introns with sizes up to thousand base pairs was also recommended to allow scalable capacity of the construct [140]. In general, it has been revealed that the choice of introns and the choice of sgRNA insertion site are both crucial for enhanced editing efficiency. Therefore with appropriate intron choice and proper positioning in the *cas* gene, a robust editing system that resembles a typical gene structure can be configured [140,141].

### 5.3. RNA Processing-Independent Configuration

Exploiting the RNA processing systems such as the exogenous Csy4 and ribozyme, and the endogenous tRNA system for the production of multiple sgRNAs from a polycistronic transcript has undeniably improved the implementation of CRISPR-Cas9-based multiplexed genome editing. The system requires specific recognition sequences for the cleavage or processing of stacked sgRNAs to occur. A more simplified format has also been reported consisting only of *cas9* and sgRNAs in a single transcript without any incorporated sgRNA processing machinery. As demonstrated by Mikami et al. 2017 [143] and Wang et al. 2018 [144], functional sgRNAs can be produced from a polycistronic transcript without the use of the previously known processing systems. The molecular construct was composed of an array of sgRNAs connected to the 3′ end of the *cas9* gene separated by 6–12 bp linkers and was under the control of a single Pol II promoter. Efficient mutagenesis of target genes in rice has been observed using the system and was comparable to the mutagenesis achieved by ribozyme-based [143] and the traditional Pol III system [144]. It has been further reported that the presence of Cas9 protein coupled with the endogenous RNases in plants is responsible for the formation of functional sgRNAs from the fused Cas9-sgRNA primary transcript. The translated Cas9 protein can be bound to the sgRNA transcript and the RNA sequences are trimmed by RNases [143]. These studies provided new insights into the understanding of sgRNA processing in eukaryotes as well as a new direction for developing simple and stable expression systems for multiplexed genome editing in plants.

## 6. Delivery System

The CRISPR-Cas9 system, like any other genome-editing tool, is a technological breakthrough that offers novel opportunities to understand fundamental insights in plant biology and commercial agriculture. Basically, for this genome editing technology to perform its intended task, it has to be introduced into the plant cell via genetic transformation (Figure 6). However, our ability to rapidly manipulate plants through genetic transformation remains a major challenge [145], thus limiting the application of CRISPR-Cas9 technology to a wide variety of plants.

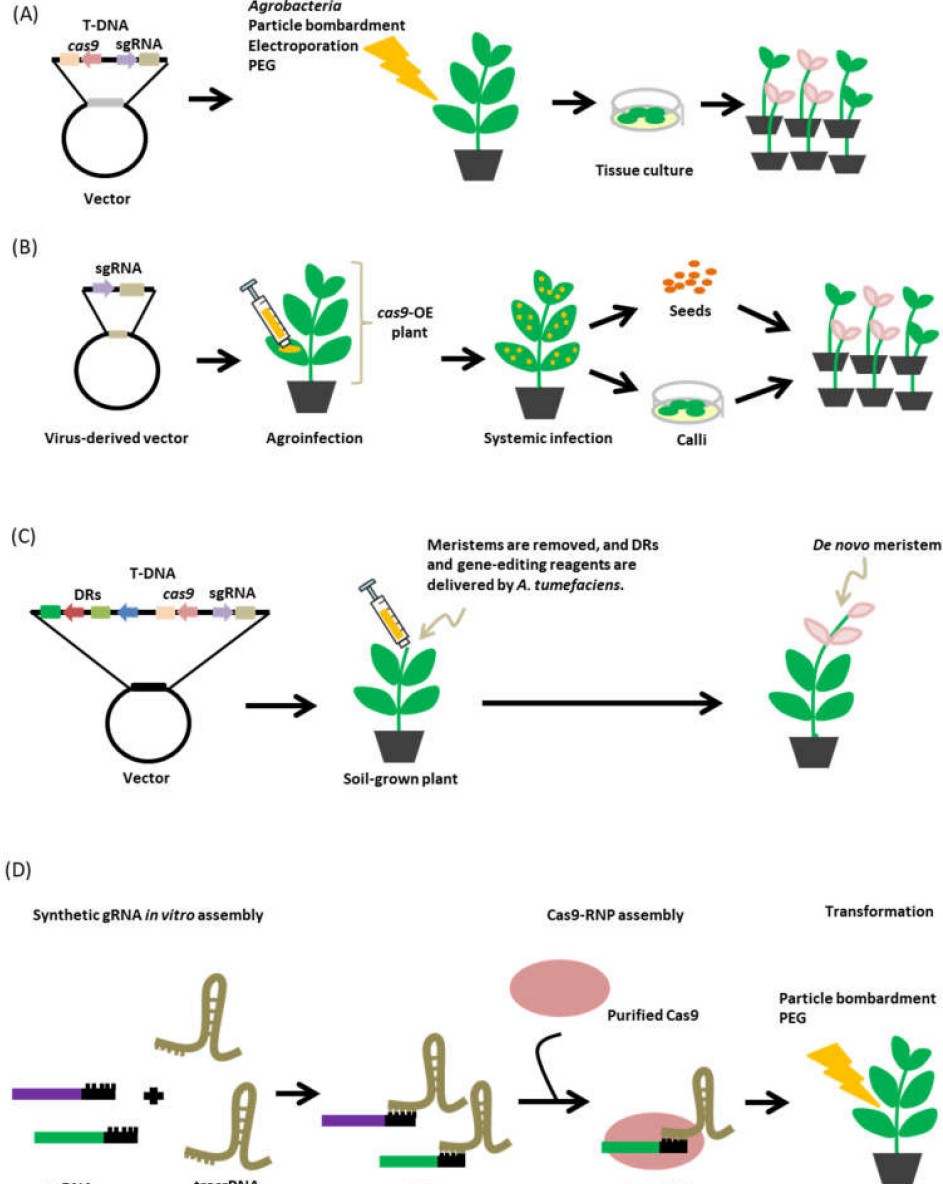

**Figure 6.** Delivery approaches. (**A**) Transformation method. Several transformation systems have been established for genetic transformation in plants. These include the *Agrobacterium*-mediated transformation, biolistic transformation of explants, and the direct transformation of protoplast with polyethylene glycol (PEG) or via electroporation. (**B**) Virus-mediated sgRNA delivery. Virus-derived vectors are used to deliver sgRNAs in plants. These vectors are introduced via *Agrobacterium* infiltration in Cas9-overexpressing (Cas9-OE) transgenic plants. The infected cells will serve as a reservoir for further replication of viral genes as well as for the systemic infection or spread into different cells or tissues including germline cells. Explants or seeds can be obtained from the infected plant and genome-edited plants can be screened. (**C**) *De novo* meristem induction. Developmental regulators (DRs) are co-expressed along with the CRISPR-Cas9 editing reagents. The meristem in the host soil-grown plant is removed. The cut site is then perfused with *A. tumefaciens* cultures carrying the vector. Transgenic meristem/shoot is produced in the site of delivery over time. (**D**) Cas9-sgRNA Ribonucleoproteins (RNPs). The formation of functional effector complex consisting of the sgRNA and Cas9 nuclease is achieved in vitro through the hybridization of synthetic crRNAs and tracrRNAs, followed by the association of the hybridized sgRNAs with a purified Cas9 protein. This in vitro assembled effector complex, capable of targeting and degrading DNA is then delivered to the host via traditional transformation method.

### 6.1. Transformation Method

Several transformation systems have been established for genetic transformation in plants such as the *Agrobacterium*-mediated and biolistic transformation of explants, and the direct transformation of protoplast with polyethylene glycol (PEG) or via electroporation [146]. These transformation methods have been proven to effectively transform plants; however, the efficiency varies and usually dependent on the plant species, and in some instances, the plants are recalcitrant or difficult to transform using these methods. As with conventional plant transgenesis, the delivery of CRISPR-Cas9 reagents in plants is achieved using these transformation methods and have been proven to successfully deliver the editing system in various plant species with varying transformation efficiencies [8]. For instance, in barley (*Hordeum vulgare* cv. "Golden Promise") higher numbers of transformants carrying the CRISPR-Cas9 reagents were observed when the genetic transformation was carried out using *A. tumefaciens* compared to those transformants produced via particle bombardment. The same trend was also observed for the frequency of CRISPR-Cas9-mediated mutations generated by the two delivery methods [147]. In wheat, on the other hand, the low efficiency of transformation methods has been a known hurdle. Certain factors must be taken into account such as the choice of genotype, quality of the immature embryo, composition of the media, *A. tumefaciens* strain, pre-treatment of the embryos, and tissue handling to achieve efficient transformation. Optimization of these factors in wheat has been shown to generate as high as 90% transformation efficiency using the *Agrobacterium*-mediated genetic transformation method [148]. Particle bombardment of wheat immature embryo and callus cells for the delivery of CRISPR-Cas9 DNA has also achieved remarkable success in obtaining transient expression of CRISPR-Cas9 reagents and transgene-free homozygous transformed plants [149]. Other crops like rice and maize, the genetic transformation is routinely achieved both through *Agrobacterium* or particle bombardment, however, the frequency of obtaining transformants greatly depends on the plant genotype. In the case of CRISPR-Cas9-mediated gene replacement, where DNA templates are to be delivered simultaneously with the CRISPR-Cas9 reagents, the biolistic transformation method is usually preferred over *Agrobacterium*-mediated transformation. The biolistic method has the advantage of delivering multiple copies of donor DNA templates into the host increasing the chance of gene replacement through homologous recombination [103,150]. Another method like the use of PEG for protoplast transformation has also proven its efficiency in transforming grapevine and apple protoplasts with CRISPR-Cas9 RNPs with a higher yield of transformants and mutation efficiency. This was achieved through the optimization of previously known protoplast transformation protocol for grapevine and apple. Factors such as the concentration of cell wall digestion enzymes, buffers, the osmotic environment of the protoplasts, incubation period, and the type of explants for protoplast isolation have been standardized for improved protoplast viability, yield, and transformation efficiency [151].

Although these common transformation protocols have decent efficiencies, their portability remains limited to certain plant genotypes. Recently, an overexpression approach of maize morphogenic regulator genes *Baby boom* (*Bbm*) and *Wuschel2* (*Wus2*) has overcome this limitation for *Agrobacterium*-mediated transformation. The overexpression approach of these genes significantly augmented the *Agrobacterium*-mediated transformation efficiency in monocots particularly in marginally transformable maize, rice, sorghum, and sugarcane varieties. The approach further enabled direct *Agrobacterium*-mediated transformation of maize mature seed embryo axes or leaf segments without undergoing callus or meristem culture step [152]. This approach might also help improve the transformation efficiency in dicots, although reports on the overexpression of these genes and other morphogenic regulators in dicots were restricted only on the regeneration or growth stimulation of embryonic tissue and none reported about transformation frequency [153–156]. Nonetheless, improving the regeneration process, a critical step in the transformation protocol will somehow increase the likelihood of generating transformants.

### 6.2. Virus-Mediated sgRNA Delivery

An alternative to the conventional approaches to delivering genome-editing reagents in plants, virus-based vectors have emerged as an efficient delivery platform for sgRNAs. The virus-mediated

sgRNA delivery system offers several advantages compared to the traditional delivery methods: (1) high levels of sgRNAs can be attained owing to the autonomous viral replication and systemic movement of the viral particle in the plants, (2) shortens and simplifies the period of operation such as the laborious transformation and regeneration process, (3) can be used for *in planta* transformation, and (4) may offer transgene-free edited plants [157–159]. Several plant RNA and DNA viruses have been engineered and reported to facilitate genome editing in plants. RNA viruses such as tobacco rattle virus (TRV) [159], tobacco mosaic virus (TMV) [160], pea early browning virus (PEBV) [161], and beet necrotic yellow vein virus (BNYVV) [162] have been used as vectors for the delivery of sgRNAs in *N. benthamiana*, *Arabidopsis*, or *Beta macrocarpa* plants. Additionally, in wheat and maize, the RNA virus barley stripe mosaic virus (BSMV) has been also exploited as a vector for the editing reagents [158]. The cabbage leaf curl (CaLCuV) geminivirus, a plant DNA virus has also been shown to efficiently deliver sgRNAs in *N. benthamiana* [163]. Other geminivirus viral replicons derived from bean yellow dwarf virus (BeYDV), wheat dwarf virus (WDV), and tomato leaf curl virus (ToLCV) have been engineered for gene targeting in potato [164], wheat [165], rice [166], and tomato [167]. The virus-derived vectors have been shown to be also efficient to deliver multiple sgRNAs for multiplexed genome editing [160,165]. In most cases, these virus-derived vectors are introduced via *Agrobacterium* infiltration in Cas9-overexpressing transgenic plants. Once introduced into the plant cells, the virus expression system facilitates the synthesis of viral genes. The infected cells will serve as a reservoir for further replication of viral genes as well as for the systemic infection or spread into different cells or tissues including germline cells [157,161,168]. This approach can provide a tissue culture-dependent or independent generation of genome-edited plants.

Virus-mediated delivery of genome editing reagents could potentially generate transgene-free edited varieties, especially with the use of RNA viruses. RNA viruses do not integrate into the plant genome, unlike DNA viruses thereby avoiding unwanted genome integration [157,165]. However, the cargo capacity of RNA viruses is limited, restricting their use for relatively small sequences in contrast to the DNA viruses [164]. It can be noted, that the induction of genome editing through this approach has been only achieved using transgenic plants overexpressing the *cas9* gene. This is mainly due to the large coding sequence of the *SpCas9* gene, hindering its incorporation especially into RNA viral vectors. On the contrary, DNA viruses like geminiviruses have larger genome sizes and could potentially carry large heterologous sequences like *cas9* gene, however cell-to-cell movement or systemic infection could be compromised as demonstrated by previous studies [169,170]. For efficient transgene-free genome editing in plants, further optimization of this delivery approach must be carried out. Engineering the system to allow transient delivery of both *cas9* and sgRNAs is thus required to facilitate the introduction of novel traits in a non-transgenic fashion. The use of Cas9 orthologues with smaller coding sequences could potentially circumvent the limited cargo capacity of viral vectors and boost the usability of the delivery system for CRISPR-Cas9-mediated genome editing in plants.

### 6.3. De Novo Meristem Induction

The development of transgenic and genome-edited plants is currently challenged by our existing tissue culture techniques, amendable only to a handful of plant species [171]. Recent approaches geared towards improving tissue culture include the use of developmental regulators (DR) that dictate in part the development or identity of the meristem. Exploiting the totipotent feature of plant cells, ectopic expression of specific combinations of DR in somatic cells has allowed potential induction of meristems as observed in *Arabidopsis* when the DRs *Wuschel* (*WUS*), *shoot meristemless* (*STM*), and a variant of *monopteros* (*MP*) were expressed in leaf cells [172,173]. A recent study by Maher et al. 2020 [174] reported the use of de novo meristem induction as a method to generate CRISPR-Cas9 genome-edited dicots plants, bypassing the necessity for tissue culture.

The approach was achieved through the ectopic co-expression of different combinations of DRs *Wuschel2* (*Wus2*) from maize, *STM* from *Arabidopsis*, together with the cytokinin biosynthesis gene

isopentenyl transferase (*ipt*). The transgene construct was delivered into *N. benthamiana* seedlings germinated in liquid cultures using *A. tumefaciens*. The different DR combinations resulted in the formation of callus-like growths localized in the regions with high expression of DRs, which eventually progressed into stems with leaflets. Two combinations of DRs, *Wus2* and *STM*, and *Wus2* and *ipt*, have been shown to result in the formation of numerous shoot-like growths. De novo meristem induction has also been observed when soil-grown plants were transformed. For CRISPR-Cas9-mediated genome editing, transgenic seedlings and soil-grown plants constitutively expressing Cas9 were transformed by the right combination of DRs and sgRNAs targeting phytoene desaturase (*PDS*) gene. Inactivation of the *PDS* gene will result in chlorophyll photobleaching, permitting easy identification of the mutant phenotype. Efficient genome editing was observed, comparable to those results obtained by the typical CRISPR-Cas9 delivery approach. The results sufficiently suggested that shoots with targeted gene edits can be generated on both seedlings and soil-grown plants through the use of DRs in combination with CRISPR-Cas9 editing reagents. The mutations generated were stable and transmission to the next generation has been observed. Moreover, many gene-edited shoots were devoid of the transgenes, cutting the need for further segregation in the next generation and could further accelerate the development of transgene-free lines for commercial applications [174].

Transgenic shoots have also been generated in some important agronomic plant species such as tomato, potato, and grape using the method implying the utility of the method in various dicot species. The functionality of the approach in soil-grown plants eliminated the need for aseptic culture and substantially trimmed down the time required for the traditional tissue culture method [174]. The use of DRs to generate de novo gene-edited meristem is indeed a promising approach that can be adopted to eventually extend in planta transformation to a variety of plant species for the rapid production of genome-edited plants.

### 6.4. Cas9-sgRNA Ribonucleoproteins (RNPs)

The functionality of plasmid or DNA-based CRISPR-Cas9 systems in plant genome editing has been proven in various studies. However, the implementation of this DNA-based system in plants requires the characterization and construction of species-specific expression cassettes and vectors, which are often laborious and sophisticated. Additionally, the use of plasmids in this system poses the risk of integrating unwanted recombinant DNA sequences into the genome of the host plant, resulting in the generation of a transgenic plant, which warrants regulatory protocols [175] and thus complicating the path to commercialization of improved varieties. To overcome these downsides, a DNA-free system has been recently established, which consisted of an in vitro-assembled Cas9-gRNA ribonucleoproteins (RNPs). In this system, *cas9* and gRNA expression cassettes are replaced by purified Cas9 nuclease and hybridized synthetic gRNA, which are then mixed in vitro to form functional RNPs, and are delivered into the host using conventional transformation methods. The delivery of pre-assembled Cas9-gRNA RNPs into the host could eliminate the risk of integrating foreign DNA in the host genome as compared to their plasmid-encoded counterparts, and could further eliminate the need for backcrossing and extensive screening of progenies. This system has been initially implemented for genome editing of human cell lines [176] and nematode *Caenorhabditis elegans* [177] and has adopted recently for genome modifications in various organisms including plants. So far, enumerable studies have been reported to have successfully used the DNA-free CRISPR-Cas9-based system in various plant species such as *A. thaliana*, *Lactuca sativa*, *N. attenuate*, *O. sativa* [175], *Z. mays* [178], petunia [179], grapevine and apple [151], *Triticum aestivum* [180,181], and *Solanum tuberosum* [182].

Aside from eliminating the possibility of transgene integration in the generated mutant, the relatively short persistence of this editing system in the host also offers additional advantages such as the absence of off-target effects [175,178,181–183]. The absence of non-specific DNA cleavage is postulated to be due to the avoidance of constitutively active CRISPR-Cas9 reagents, shortened functional time, and fast cellular degradation of Cas9-gRNA RNPs in the host [181]. As demonstrated in human cells, Cas9-gRNA RNPs were degraded by endogenous proteases in the

host after cleavage of chromosomal target sites [176]. It has been indeed agreed that decreasing the functional time of CRISPR-Cas9 reagents will result in a concomitant decrease in off-target effects. The lack of DNA construct in this DNA-free system is also an added advantage as there is no need for expression cassettes or plasmid vector construction and other things like codon usage optimization and finding promoter to efficiently drive the expression of *cas9* and gRNAs. Thus, the system can be readily used without intricate adaption procedures for various organisms. The use of pre-assembled Cas9-gRNA RNPs can substantially broaden the applicability or portability of the genome editing system across different genetic backgrounds of all transformable plant species.

Although the DNA-free system directs a new avenue for transgene-free genome editing through CRISPR-Cas9, the system is still troubled by certain drawbacks. The editing efficiency is lower compared to the classical DNA-based method, and the selection process is difficult after transformation because no selection marker is present and if not through phenotype screening, the validation of successfully edited plants is solely possible through molecular approaches [184].

## 7. Detection Method for Transgene-Free Edited Plants

After successful delivery of CRISPR-Cas9 editing reagents, isolation of the transformants with the expected mutations is the primary task. The screening of the primary edited plants would be easy if the mutation confers striking phenotypic evidence. If not, genetic characterization is usually performed like restriction enzyme digestion or sequencing of amplicons [185–187], which typically consume a lot of time as well as resources. Isolating transformants and verifying the presence of intended mutations are not just the end tasks, especially for high valued crops. The CRISPR-Cas9 construct must be subsequently eliminated. The presence of the CRISPR-Cas9 construct in the genome of edited plants makes it difficult to distinguish the transmission of mutations from generations as new mutations can arise in every generation, hampering the full assessment of genetic heritability. Moreover, the prolonged existence of the editing construct could potentially cause off-target effects, making phenotypic stability a concern. Further, for commercial application of genome-edited plants transgene-free is likely a prerequisite for regulatory approval [188]. Eliminating the CRISPR-Cas9 construct in the genome of edited plants is usually achieved by a repeated self-crossing or backcrossing. However, neither of these strategies is short nor simple. The quick elimination of transgene and screening of transgene-free edited plants remains a challenging part after genome editing [189]. To this end, a few different methods have been established to improve the efficiency of screening or isolation of transgene-free genome-edited mutants (Figure 7). These methods include the fluorescence marker-assisted selection [135,185,189,190], active interference element-mediated selection [191], programmed self-elimination system [188], bolting-assisted selection [192], and $H_2O_2$-based leaf painting assay [193].

### 7.1. Fluorescence Marker-Assisted Selection

Like antibiotic and herbicide resistance, fluorescent proteins have also been conventionally used as markers for transgene integration in plants. The fluorescent proteins provide a rapid screening process of stably transformed plants without affecting the plant viability [194,195]. Basing on the fluorescent signal "on" or "off", potential genome-edited plants without the integrated CRISPR-Cas9 editing reagents can be sufficiently identified. For instance, the use of *mCherry* expression cassette integrated into the CRISPR-Cas9 vector has visually differentiated the seeds harvested from *Arabidopsis* T2 plants into two groups: a group displaying strong red fluorescence and another group with no fluorescence. Because both the *mCherry* cassette and CRISPR-Cas9 unit are in one molecular construct, the lack of red fluorescence indicates a transgene-free state. Genetic characterization of the plants derived from the non-fluorescent seeds revealed desired mutations without the CRISPR-Cas9 construct, suggesting the reliability of the fluorescence-based method [135,190]. This selection method has been adopted and applied in the CRISPR-Cas9-mediated genome editing of a wide range of plant species in addition to *Arabidopsis*, such as *Brassica napus*, *Fragaria vesca* (strawberry), and *Glycine max* (soybean) using green fluorescent protein (GFP) as a visual marker. GFP-positive signals were readily

identified in transformed plants and the mutation efficiency resembles that of the original vector. Moreover, no off-target mutations have been identified among potential off-target loci. Based on the absence of GFP fluorescence, transgene-free mutants of *Arabidopsis* and *B. napus* have been sufficiently identified in T2 and T1 generations, respectively, and mutations were stably transmitted to the next generations [185]. Similarly, Aliaga-Franco et al. 2019 [189] demonstrated the use of DsRED fluorescence-based visualization in tomato, rice, and *A. thaliana* for the detection of primary transformants and CRISPR-Cas9 construct-free mutants. Transgene-free DsRED-negative plants with expected gene mutations were isolated in a single generation after in vitro transformation. Mutations in the transgene-free edited plants were stably inherited in the next generations as well. Taken together, these studies established an effective tool to readily identify positive primary transformants and edited plants devoid of CRISPR-Cas9 construct, which can be extended to a wide range of dicot and monocot plant species. Of note, fluorescence-based screening must be combined with genetic analysis to ensure the loss of the integrated transgene. Nonetheless, the method offers a reliable screening and would speed up the isolation of transgene-free CRISPR-Cas9-edited plants.

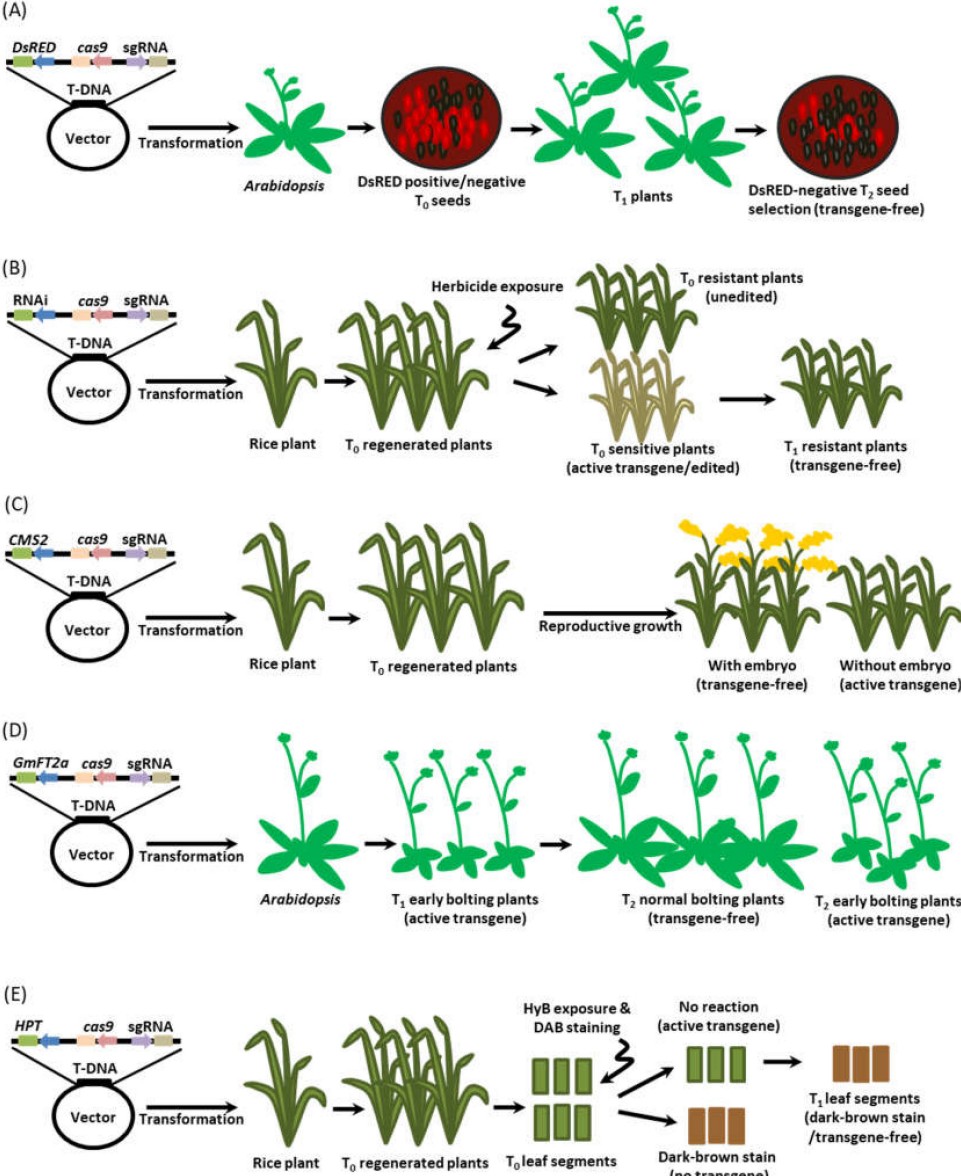

**Figure 7.** Detection methods for transgene-free edited plants. (**A**) Fluorescence marker-assisted selection. Expression cassettes of fluorescent proteins (e.g., GFP, mCherry, and DsRED) are incorporated in the

CRISPR-Cas9 vector. Following transformation and subsequent regeneration, seeds can be harvested from plants and fluorescent signals can be observed using a fluorescence microscope. Based on the fluorescent signal, potential genome-edited plants without the integrated transgenes can be sufficiently identified. (**B**) Active interference element-mediated selection. Silencing genes coding for a striking phenotype through RNAi can be used as a selection marker for the isolation of transgene-free edited plants. Incorporating the RNAi expression element in the CRISPR-Cas9 vector allows the monitoring of the activity and the presence of T-DNA. For instance, incorporating an RNAi against the herbicide resistance gene in rice permits the identification of transgene-free edited plants in the subsequent generations following herbicide exposure. Rice plants showing sensitivity to the herbicide denote an active RNAi and the presence of the transgene. Resistant plants are thus considered transgene-free edited plants. (**C**) Programmed self-elimination. Plants are transformed with vectors containing a T-DNA with suicide genes (e.g., *BARNASE* and *CMS2*) and CRISPR-Cas9 cassettes. These suicide genes are activated during the reproductive growth of the plants inhibiting the development of the embryo. The temporal control of their expression allows ample time for CRISPR-Cas9 reagents to perform genome editing. Plants with developed embryo during the reproductive growth are devoid of transgenes. (**D**) Bolting- assisted selection. Bolting or flowering is a striking phenotype that can aide in the selection of transgene-free edited plants. Incorporating a gene that positively regulates flowering (e.g., *GmFT2a*) in the CRISPR-Cas9 construct would provide a method for the easy identification of genome-edited transgene-free mutants. Early bolting plants obtained following transformation indicate the presence of transgenes. Transgene-free genome-edited plants can then be obtained by selecting plants exhibiting normal bolting phenotype in the succeeding generations. (**E**) $H_2O_2$- based leaf painting. The method is based on HyB-induced $H_2O_2$ accumulation in rice plants, which can be detected by staining with DAB. DAB, when taken up by plants, reacts with $H_2O_2$ forming a dark-brown reaction product, thus providing a visual feature that can be used for screening. In this method, the *HyB* gene is incorporated in the CRISPR-Cas9 vector and introduced into the plant through genetic transformation. The presence of *HyB* renders the plant resistant to the HyB-induced $H_2O_2$ accumulation. Leaf segments from the regenerated plants are then obtained and treated with HyB and subsequently stained by DAB. Plants that containing the transgene will show no reaction with the stain, while transgene-free edited plants will exhibit dark-brown color.

### 7.2. Active Interference Element-Mediated Selection

*Agrobacterium* T-DNA is typically used as a vector for introducing genes into the genome of plants. This engineered vector, carrying the genome-editing reagents like the CRISPR-Cas9 system will be randomly inserted into the plant genome. Due to this random insertion nature of the vector, the T-DNA may be silenced especially when inserted in heterochromatin or is actively silenced by the host silencing mechanisms [196]. It could be postulated that only a portion of transformed plants may carry active transgenes and hence, the screening of $T_0$ genome-edited plants and subsequent selection of transgene-free plants would be challenging.

RNA interference (RNAi) is an established technique used to silence genes. The silencing phenomenon involves sequence-specific gene regulation induced by double-stranded RNA (dsRNA), resulting in the inhibition of transcription or neutralization of mRNA, thus inhibiting translation [197]. This technique has been applied for improvements in various crops. Due to the dominant nature of functional RNAi like hairpin RNAi (hpRNAi), their silencing activity can be monitored in the transformed $T_0$ plants. Backup with this fact, incorporating an RNAi expression element in the CRISPR-Cas9 vector would allow monitoring of the activity and presence of the T-DNA in the transgenic plants. This would further enable a phenotype-based screening of genome-edited $T_0$ plants and subsequent isolation of transgene-free plants in later generations. This strategy was established by Lu et al. 2017 [191]. The group incorporated in the CRISPR-Cas9 vector an RNAi expression element that targets a gene (*CYP81A6*) that confers resistance to bentazon herbicide in rice. Susceptibility to the bentazon herbicide would indicate active T-DNA, allowing the screening of transformed and genome-edited $T_0$ plants. Mixtures of bentazon resistant and susceptible regenerated transformed

plants were obtained, suggesting a strong possibility of transgene silencing among the transformed plants. The expression of *CYP81A6-hpRNAi* resulted in the degradation of *CYP81A6* transcripts in transgene-expressing susceptible $T_0$ plants. In contrast, the transgene-silenced $T_0$ plants remained resistant. The herbicide susceptible plants were found to have the desired gene mutation, thus correlating susceptibility with the targeted mutations in $T_0$ plants and further implied an efficient screening for genome-edited $T_0$ plants. The system also simplified the selection of transgene-free $T_1$ plants by sorting out the bentazon resistant plants in which the *CYP81A6-hpRNAi* no longer existed. In general, the system has exhibited a reliable method to indirectly estimate the expression level of Cas9 editing reagents by correlating with a marker phenotype generated by an RNAi element incorporated into the CRISPR-Cas9 vector, which subsequently provided a cost-effective and simply way of eliminating and identifying $T_0$ non-edited plants and transgene-free $T_1$ plants, respectively. This system can be extended to other plant species using the homologs of *CYP81A6* gene or other marker traits that introduce visible phenotypic changes.

### 7.3. Programmed Self-Elimination System

Another strategy developed for efficient and fast isolation of transgene-free CRISPR-Cas9-edited plants is the so-called Transgene Killer CRISPR (TKC) technology [188]. The method is employing a pair of suicide transgenes that effectively kill and eliminate CRISPR-Cas9-containing pollen and embryos produced by $T_0$ rice plants. The system utilized the bacterial *BARNASE* gene [198] that encodes a toxic protein known to kill plant cells, and *CMS2* gene that encodes the rice male gametophyte specific lethal protein. The expression of the *BARNASE* gene was driven by the rice *REG2* promoter to achieve temporal control of the expression during embryo development. The temporal control of the suicide transgenes allows ample time for CRISPR-Cas9 construct to perform targeted mutagenesis in calli and in vegetative cells of $T_0$ plants before its removal. Upon reproductive growth, both the incorporated *BARNASE* and *CMS2* cassettes will produce toxic proteins that kill male gametophytes and embryos that contain the CRISPR-Cas9 construct, respectively, resulting in the self-elimination of the system. Seeds obtained from the $T_0$ plants are thus transgene-free with the desired mutation. A 100% efficacy was observed in inducing the target mutation as well as in eliminating the transgenes in $T_1$ rice plants, demonstrating the robustness of the system.

The use of the system demonstrated the efficacy of the suicide transgenes *BARNASE* and *CMS2* in producing transgene-free rice plants in a single generation without compromising the editing efficiency of CRISPR-Cas9. Further, the system greatly reduced the required time and labor for isolating transgene-free CRISPR-Cas9-edited rice plants.

### 7.4. Bolting-Assisted Selection

Flowering is a striking phenotype that can be helpful for the screening process. In most flowering plants, appropriate flowering time is considered critical for successful sexual reproduction. In the model plant *Arabidopsis*, the flowing time is controlled by several different regulators that sense and respond to environmental signals. Among the different flowering regulators, the *flowering locus T* (*FT*) is the key positive regulator of flowering in *Arabidopsis* [199–202]. Overexpression of *FT* has been successfully used to accelerate the process of plant breeding by reducing the juvenile phase of many plants [203–205]. Considering the central role of *FT* in flowering, it may serve as a selection marker that can assist easy identification of transgene-free mutants. Moreover, shortening the juvenile phase by early flowering due to the overexpression of *FT* may reduce the overall time required for generating genome-edited mutants. Thus, incorporating *FT* in the CRISPR-Cas9 construct would provide a method for fast generation and easy selection of genome-edited transgene-free mutants. As demonstrated by Cheng et al. 2019 [192], incorporating a *35S* promoter-driven *GmFT2a* cassette in the CRISPR-Cas9 construct resulted in an early bolting phenotype of *Arabidopsis* transgenic $T_1$ plants. Bolting time of the transgenic plants was observed as early as 8 days whereas the wild type plants ranged from 17 to 20 days. Genotypic validation of the mutation in the target sites in early bolting $T_1$

plants showed efficient mutagenesis, hence correlating the presence of transgene with the early bolting phenotype. Transgene-free genome-edited plants were obtained in $T_2$ exhibited by the normal bolting time phenotype. Aside from the easy identification of transgenic and transgene-free mutants based on phenotypic assessment, the early bolting phenotype also hastened the duration for generating genome edited transgene-free plants by reducing the length of the juvenile phase. The overall selection process, starting from the identification of transgenic $T_1$ plants up to the final identification of transgene-free genome edited $T_2$ plants, has been reduced by at least 10 days.

The presence of *FT* homologs in other plant species [203–205] (e.g., medicago, rice, soybean, and trees like poplar and pear) could imply the possible adoption of this method for easy screening and fast generation of transgene-free CRISPR-Cas9-edited plants. Genome editing-based breeding for plants with a long juvenile phase can also possibly benefit from this method [192].

### 7.5. $H_2O_2$-Based Leaf Painting

Leaf painting assay is a method used to facilitate the screening of transgenic plants. The method is based on the tolerance of transgenic plants to antibiotics or herbicides, which can be visually observed in the leaf. The process usually involves the integration of antibiotic or herbicide resistance cassette in the transgene vector construct and the transgenic plants are screened by painting the leaf with the antibiotic or herbicide and are then observed for wilting symptoms. Leaves showing the wilting phenotypes are considered transgene-free plants while those with no wilting symptoms contain the transgene. This method has been successfully implemented to screen transgenic lines of cotton [206], rice [207], and maize [208]. Utilizing this method for the isolation of transgene-free CRISPR-Cas9-edited plants would be an addition to the number of methods recently developed, however, the typical leaf painting assay usually takes time as visible leaf wilt symptoms appear almost one week. Developing an efficient and rapid leaf painting assay is thus needed for high throughput screening of transgenic or transgene-free genome-edited plants.

Recently, Wu et al. 2019 [193] developed a leaf painting assay based on hygromycin B (HyB)-induced $H_2O_2$ accumulation for the detection of transgene-free CRISPR-Cas9-edited rice. It has been reported that exposure to HyB can significantly enhance the production and accumulation of $H_2O_2$ in rice leaves [171]. The accumulation of $H_2O_2$ can be detected by staining with 3,3-diaminobenzidine (DAB). DAB, when taken up by plants, reacts with $H_2O_2$, forming a dark-brown reaction product in the presence of peroxidase, thus providing a visual feature that can be used for the screening of transgenic and transgene-free plants. Anchored on this rationale, the hygromycin phosphotransferase gene (*HPT*) that confers resistance to HyB was incorporated in the CRISPR-Cas9 vector. Transgenic $T_0$ rice plants were sufficiently identified after exposure of the leaf segments to HyB and subsequent DAB staining. The non-transgenic rice plants showed the dark-brown reaction product on the leaves suggesting the accumulation of $H_2O_2$ upon exposure to HyB. Genome editing events mediated by CRISPR-Cas9 were verified in the transgenic $T_0$ rice plants by genotyping, suggesting the reliability of the method for detecting transgenic plants. The transgene-free edited mutants were obtained in $T_1$ rice plants showing no visible $H_2O_2$ accumulation on the leaves after exposure to HyB. The method reduced the length of time required for a typical leaf painting assay to reveal significantly phenotypic result in rice. Instead of the usual five to seven days, the recent method could show result in just 12 h. Further, $H_2O_2$ production upon exposure to HyB has been observed in other plants such as *Arabidopsis*, tobacco, tomato, and maize, suggesting that the $H_2O_2$-based leaf painting assay can be established in a wide variety of plants, including both dicot and monocots for a reliable and rapid screening of transgene-free CRISPR-Cas9 genome edited plants.

## 8. Improved CRISPR-Cas9 Efficiency by RNA-Silencing Inhibition and Heat Stress

As a powerful genome engineering tool, the CRISPR-Cas9 system has received so much attention regarding its editing efficiency in various biological systems. It has been a known fact that the editing efficiency of the system varies greatly in different organisms; as such various studies have surfaced

focusing on the improvement of the system's editing efficiency. Most of these studies have centered the improvement at the transcriptional level such as the characterization of vectors and promoters to drive the expression of *cas9* and sgRNA cassettes, and have shown considerable improvement on the editing efficiency of the system. Although many of these plasmid-based systems have shown efficient targeted gene editing in plants, the high transgene expression levels from the editing system potentiate the activation of transgene-silencing machinery of the host thereby interfering in the editing activity. Recent studies have demonstrated the involvement of host post-transcriptional regulation, specifically the RNA-silencing pathway on the editing efficiency of CRISPR-Cas9 in plants [209,210]. Inactivation of the genes involved in the RNA silencing pathway in *Arabidopsis* resulted in higher mutagenesis frequency mediated by CRISPR-Cas9 compared to the wild type plant. Increased in the transcript levels for both sgRNAs and *cas9* have also been observed, linking a strong correlation between expression levels and mutation efficiency [210]. In addition, expression of the viral suppressor p19, an RNA-silencing suppressor protein derived from tomato bushy stunt virus (TBSV) [211] in *Arabidopsis* significantly increased the editing frequency of CRISPR-Cas9 with a concomitant increase in sgRNAs and *cas9* transcripts as well [212].

Optimum growth temperature requirement varies in different biological systems. Temperature is known to affect many biological processes such as enzyme kinetics, chromatin structure, DNA repair pathways [213,214], and even could directly affect the efficiency of CRISPR-Cas9-targeted mutations in the eukaryotic genome. A study revealed the importance of temperature in modulating the activity of SpCas9 [214]. It has been demonstrated that exposing *Arabidopsis* and citrus plants to repeated heat stress at 37 °C growth temperature showed much higher frequencies of CRISPR-Cas9-induced mutations compared to plants grown at 22 °C standard temperature. Quantification of the green fluorescent protein (GFP) reporter gene has further validated the increased targeted mutation in *Arabidopsis* somatic tissues (5-fold) and up to a 100-fold increase in germline upon heat exposure. The increased editing efficiency of CRISPR-Cas9 has been attributed to the activity of SpCas9 nuclease at 37 °C, as the in vitro assay revealed robust activity of SpCas9 in inducing DSBs at 37 °C than at 22 °C. Thus, the observed variation in the efficiency of CRISPR-Cas9-mediated mutations in different organisms can possibly be contributed to by the variations in the growth temperature.

## 9. Summary and Concluding Remarks

The CRISPR-Cas9 system is a simple and straightforward genome editing tool that considerably revolutionized our ability to perform targeted genome modifications in various organisms. This novel editing platform has superseded the early known editing tools due to its simple design of targeting vector, high success rate, and low cost. In plants, the system has been successfully utilized for various applications including the generation of improved plant varieties. Despite its superior and proven ability to carry out targeted genome modifications, the application of CRISPR-Cas9 in plants is still troubled by certain limitations and regulatory concerns. Various parameters, together with the existing approaches, are to be considered and optimized for the efficient delivery and accurate targeting of the system. Like any other genome-editing tool, the off-target effect is one of the major concerns as this may lead to unwanted mutations. In this regard, efforts have been rendered to characterize the CRISPR-Cas9 editing reagents for the development of approaches that can carry out precisely tailored mutations with reduced incidents of off-target events. These efforts have led to the development of online bioinformatics tools that assist in the selection of highly specific sgRNAs. In addition, high-fidelity variants of the Cas9 nuclease and its orthologues have been developed and exploited, respectively to minimize off-target effects and to cater to a broad range of PAMs for expanded gene targeting. The extensive characterization of decent promoters for both *cas9* and sgRNA cassettes has provided vivid and wider options for constitutive and spatiotemporal control of the gene targeting reagents, resulting in an enhanced and precise genome editing. The editing capability of CRISPR-Cas9 reagents is variable depending on the combinations of sgRNAs and *cas9* expression cassette or the entire expression vector. Indeed, various developments in expression vector architecture have been

established that have improved the targeting efficiency of the system and also simplified the vector construction process. Advancements in multiplexing strategies through the exploitation of different processing mechanisms for precise and efficient production of multiple sgRNAs have considerably eased the implementation of the multiplex editing system without compromising the efficiency of mutagenesis. As mentioned, the efficiency of CRISPR-Cas9 is variable especially from one plant species to another. Hence, fine-tuning of the conditions for its utilization in a particular species is needed to ensure successful and highly efficient genome editing.

Like the molecular components of the CRISPR-Cas9 system, delivering the CRISPR-Cas9 tools is crucial for achieving efficiently targeted genome mutation. As with traditional transgenesis, the same established genetic transformation approaches have been used for the introduction of CRISPR-Cas9 reagents in plants. However, these approaches are amendable only to a number of plant species and the transformation efficiencies greatly vary, depending on the plant genotype. In most instances, the regeneration or tissue culture step in the transformation protocol is highly problematic and labor-intensive in some plant species. Current developments in this aspect involve the overexpression of morphogenic/developmental regulator genes for improved regeneration ability, and for de novo induction of meristem in grown plants. The latter approach has condensed the overall transgenesis protocol by obviating the need for traditional tissue culture techniques. The use of virus-derived vectors for the delivery of CRISPR-Cas9 editing reagents has also provided options for tissue culture-dependent or independent transformation. These advancements further hold a promising avenue for the extended adoption of a wide variety of plants and the rapid generation of transgene-free edited plants.

Successful implementation of CRISPR-Cas9-mediated genome editing in crops usually results in the generation of transgenic lines brought by the stable integration of heritable editing reagents. A recent development in the CRISPR-Cas9 system has introduced the pre-assembled RNP approach, eliminating the need for DNA-based vectors. The RNP approach facilitates transient genome editing events, without producing transgenic plants. However, the efficiency of mutagenesis is lower compared to the classical DNA-based counterpart. Thus far, the delivery of RNPs is achieved through the biolistic bombardment of immature embryos or PEG-mediated transformation of protoplasts, which limits the portability of the approach in plants with established tissue culture techniques. Screening of successful primary transformants is difficult also due to the lack of selection marker affiliated with the method. Further improvements of this approach are needed to uplift its editing efficiency and portability for a greater chance of generating transgene-free genome-edited plants.

Ensuring the absence of transgenes—the CRISPR-Cas9 editing construct, in all genome-edited plants is vital for regulatory approval of improved varieties. Besides, further genome editing events will be avoided and genetic heritability will be properly assessed. Eliminating the integrated transgenes in the genome of edited plants requires subsequent genetic segregation, and further validation through genotyping. The approach is valid; however rapid screening of transgene and transgene-free mutant plants is hindered by the lack of striking phenotypes for differentiation, thus requiring extensive genotyping during the early screening process. Recent advancements have geared towards designing methods focused on the visual screening process for easy and rapid differentiation of transgene and transgene-free edited plants. These include the use of fluorescent proteins, interference elements (RNAi) for concomitant phenotype knockout, flower and seed-based detection, and leaf/chemical-based detection method. Each of these methods have the potential to shorten the screening and the overall process for isolating transgene-free genome-edited plants. The adoption of the method across different plant species is also possible as the design of the construct is simple and can be readily adjusted to fit with the condition of the desired host plant.

While improvements of the CRISPR-Cas9 system on the transcriptional level have remarkably enhanced its targeting scope, fidelity, and specificity; altering the RNA-silencing mechanisms of plants has also shown to augment the efficiency of the system. Strong expression activities of transgenes are usually subjected to silencing through this system. It is thus noteworthy to carefully assess vectors regarding their expression patterns for optimum expression of the editing reagents, resulting in an

efficient genome editing. An interesting finding regarding the effect of heat stress on the editing efficiency of CRISPR-Cas9 has also opened a new track to be exploited for further optimization approaches.

In general, the implementation of the CRISPR-Cas9 technology in plants is constantly in progress as researchers seek to improve and address the limitations of the system. The technology has also extended its applicability beyond genome editing [215,216]. Currently, several approaches have been detailed to provide options and to guide researchers to come up with the appropriate schemes in employing the genome-editing tool depending on the intent and nature of the research. It can also be expected that future improvements will be based on the combination of these established existing approaches. Desirably, more advancement in the future should be made tackling the delivery/transformation method of the system, as this hinders the extended application of the editing system in many plant species.

**Author Contributions:** J.A.V.M., L.L.C., and H.B. discussed the concept of the manuscript. J.A.V.M. contributed to the literature collection and writing of the manuscript. H.B. critically reviewed and revised the manuscript. All authors have read and agreed to the published version of the manuscript.

**Funding:** This work was supported by a grant from the Next-Generation Biogreen 21 Program (Project No. PJ013655022020), Rural Development Administration, Republic of Korea.

**Conflicts of Interest:** The authors declare no conflict of interest.

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
