# Peer review of "CRISPR-Cas9 System for Plant Genome Editing: Current Approaches and Emerging Developments"

_agronomy, doi:10.3390/agronomy10071033_

Round 1
Reviewer 1 Report
Montecillo et al. provided a very comprehensive review covering many aspects of CRISPR-Cas9-mediated genome editing in plants. The authors did a great job describing the current development and applications of CRISPR technologies, including currently available CRISPR reagents, expression and multiplexing systems, delivery methods and approaches to detect transgene-free edited plants. The authors also discussed some strategies to further improve the editing efficiency, such as inhibiting gene silencing and heat treatment. This review provided an informative and detailed overview of this field, which could be very useful for researchers who are interested in the CRISPR technology and its application in diverse plant species, especially those who are new to this field. This review also provided some insight into the future development of this technology. However, there are some concerns need to be addressed by the authors:
- Line 128-129: The information provided here regarding to the classification of the CRISPR systems is not up to data. Please cite the latest classification of CRISPR systems: Makarova, K.S., Wolf, Y.I., Iranzo, J. et al. Evolutionary classification of CRISPR–Cas systems: a burst of class 2 and derived variants. Nat Rev Microbiol 18, 67–83 (2020). https://doi.org/10.1038/s41579-019-0299-x
- Line 161-162: The single RNA chimera talked here is usually referred as a single guide RNA (sgRNA), while a guide RNA is usually referred as a gRNA. gRNA is also commonly used for other CRISPR systems. But gRNA does not necessarily mean sgRNA specifically used in the Cas9 system. Please keep this consistent and accurate through the entire manuscript.
- Figure 3(A) is not correct. The crRNAs are first transcribed as a CRISPR array, then hybrid with tracrRNA, followed by the cleavage of Cas9 and RNase III.
- Line 340-342: The authors stated that a high GC content is correlated with a high editing efficiency. This statement is not accurate. Editing efficiency will be decreased when the GC content is too high or too low, usually higher than 60% or lower than 40%.
- Line 376: Recently, a near-PAMless SpCas9 variant (SpRY) has been engineered with an expanded targeting scope. Although this Cas9 variant has not been demonstrated in plants, it is still a breakthrough in this field and will be soon applied in plants. Therefore, it is worth discussing here with the following citation: Walton, R.T., Christie, K.A., Whittaker, M.N., and Kleinstiver, B.P. (2020). Unconstrained genome targeting with near-PAMless engineered CRISPR-Cas9 variants. Science 368, 290–296.
- Line 433-435: Although this statement is supported by the following two citations, this statement is too conclusive. It might not absolutely true in other cases or plant species.
- Line 489: In this paragraph, the authors discussed the advantages of using PoI II promoters. It is worth mentioning that the use of PoI II promoter to drive sgRNA expression requires the sgRNA processing system. Any extra sequences in sgRNA could potentially decrease the editing efficiency or completely abolish its function, although it is less an issue when extra sequences are present at the 3’end of Cas9 sgRNA.
- Line 669: The authors stated that the tRNA processing system is precise. This might be a bit overstated, since tRNA cleavage will leave 1-4 nt extra tails at the 3’ end of the sgRNA. This does not affect the sgRNA function in the Cas9 system, but could cause problems when accurate cleavage is required, such as in the Cas12a system.
- Line 676-679: Please refer to Figure 5.
- Line 697-701: Although the use of PolyA or linker sequence instead of a terminator can assure the continuous transcription of sgRNAs, it is also worth mentioning that PolyA serves as an important structure for mRNA to potentially facilitate its exportation, translation, and stability.
- Line 733-738: Please indicate that these speculations are from the original research paper, not from this review paper.
- Line 873: The authors mentioned earlier that “the cargo capacity of RNA viruses is limited”. It is better to discuss the possibility to deliver both Cas9 and sgRNAs using virus, since Cas9 has a large size.
- Line 884-909: Please add citation for these two paragraphs. Citations usually cannot be carried over to the following paragraphs.
- Line 949: This sentence is overstated. Genome editing can also be detected using other methods than genomic sequencing, such as enzyme digestion-based methods.
- Line 950-951: This sentence is overstated. Whether someone chooses transgene-free approach or not should depends on the objective of the project.
- Line 952: It is highly recommended to add another figure for this section to show all the detection methods for transgene-free edited plants.
- Line 963-964: This statement is not accurate. Transgenic plants can also be deregulated, but usually require a lengthy and expensive process. Transgene-free edited plants can potentially not be regulated at some levels in certain countries.
- Line 1023-1025: The statement is not accurate. The mixtures of bentazon resistant and susceptible plants could also be due to that some plants are not successfully transformed. The resistant plants may contain no transgene at all, but not have silenced transgenes.
- Line 1037: This study described by Lu et al. 2017 were not able to detect silenced T-DNA. If the T-DNA is silenced in the plant genome, RNAi components can not be expressed and used for herbicide resistance selection. Therefore, edited herbicide resistant plants could potentially contain silenced T-DNA insertion. If this is the case, please reorganize this paragraph and the previous paragraph to avoid misleading.
- Line 1152: It is recommended to rewrite this section, since this section is lengthy and repeated many contents from the previous sections. Please keep it short and precise and avoid repeating what have been discussed intensively before.
- Line 1155: The word “superseded” may be overstated. Although CRISPR is the predominant approach to achieve genome editing, other technologies, such as TALEN, still have some advantages in some circumstances. For instance, TALEN is not limited by PAM sequences. In addition, a better choice for genome editing in organelles is TALEN, since directing protein into organelles is easier than directing RNA.
- Line 1166: It is worth mentioning the development of high-fidelity Cas9 variants to minimize off-target effects.
- Line 1248-1250: This study by Aschenbrenner et al. 2020 mainly focused on limiting off-target effects. This part should be moved to previous sections where off-target effect is discussed.
- Line 1259-1260: This sentence might be overstated based on a single study.
- Line 1261-1263: Since the topics regarding regulation and public education is not the focus of this review, this is not an appropriate ending for this review.
- The writing of this review can be greatly improved. It is highly recommended that the authors should let a native speaker editing this article during the revising process. It is important to keep the language concise and precise. Here are some recommendations to improve the writing:
- Line 40: Change “highlighted” to “highlights”
- Line 41: Add “is” before “favored”
- Line 43: Add “,” before “which” throughout the entire article.
- Line 46: Add “,” before “which”
- Line 55: Add “,” before “resulting”; change “resulting to” to “resulting in” throughout the entire article
- Line 69: Change “allowed” to “allows”
- Line 73: Change “has shown great advantage and more straightforward” to “has shown great advantages and is more straightforward”
- Line 74: Change “targeted-DSBs” to “targeted”
- Line 77: Add “more” before “versatile”
- Line 78: Change “rate” to “rates”
- Line 79: Change “on” to “about”
- Line 82: Add “,” before “however”. Please do this throughout the entire article
- Line 90-91: Change “straightforward and enhanced genome editing efficiency of the system” to “simple and efficient genome editing systems”
- Line 96-97: Change “involved aversion of phage attachment, blocking DNA entry” to “involve phage attachment aversion, DNA entry blockage”
- Line 104: Add “,” before “which”
- Line 118: Add “a” before “CRISPR array”
- Line 119: Change “system shows” to “systems show”
- Line 120: Add “are” before “currently classified”
- Line 143: This sentence “Following the stages of CRISPR-Cas immunity” is not clear, please rewrite it.
- Line 144: Change “the host genome, encoded as CRISPR array” to “the CRISPR array in the host genome”
- Line 151: Change “having” to “with”
- Line 162: Change “made the system simplified” to “simplified the system”
- Line 170: Change “specific DNA sequence with the recognized PAM” to “specific DNA sequences with recognizable PAMs”
- Line 170-171: Delete “, directing the nuclease activity of Cas9 to the target DNA sequence”
- Line 179-180: Change “a fusion construct of crRNA and tracrRNA which is transcribed forming a gRNA” to “a construct of a single guide RNA, which is a fusion of a crRNA and tracrRNA”
- Line 180: Add “a” before “complex”
- Line 184: Change “significant array” to “a large number of”; delete “have” before “proven”; delete “functional”
- Line 187: Change “knock out and knock in gene alterations” to gene knockout and knockin”
- Line 188: Add “,” before respectively. Please do this throughout the entire article
- Line 196: Change “to functionally characterized genes” to “characterize gene functions”
- Line 206: Change “high cost and time consuming process” to “their costly and time-consuming processes”; Change “approach” to “approaches”
- Line 207: Change “has” to “have”
- Line 217: Change “time” to “time-”
- Line 220: Add “,” before and after “respectively”
- Line 221: Change “been developed also” to “also been developed”; move “simultaneously” to the end of the sentence
- Line 224: Delete “too”
- Line 225: Add “as well” after [57]
- Line 227: Add “,” before “respectively”
- Line 228: Change “in generating” to “to generate”
- Line 253: Change “harbor mutation” to “harboring mutations”
- Line 258: Change “which” to “that”
- Line 259: Change “has been reported also” to “has also been reported”
- Line 264: Add “,” before “thereby”
- Line 265: Change “field drought condition” to “drought conditions”
- Line 293-294: Add “,” before and after “which were previously characterized based on spontaneous mutants or RNAi knockdown lines”
- Line 300: Add “the” after “paved”; change “network” to “networks”
- Line 304: Add “the” before “CRISPR”
- Line 307: Change “exerted” to “exerts”
- Line 308: Change “mutagenesis of gene” to “gene mutagenesis”
- Line 311-312: Change “to regard with in employing the technology” to “regarding the application of the technology”
- Line 318: Add “the” before “overall”
- Line 327: Change “and should be selected considering that it lies upstream of” to “followed by”
- Line 328: Add “the” before “Cas9”
- Line 329: The word “Consequently” is not used properly here
- Line 330-331: Please rewrite this sentence
- Line 335: Move “in silico” after sgRNAs
- Line 341: Change “has been shown also” to “has also been shown”
- Line 355: Change “particular concern specially” to “a particular concern especially”
- Line 356: Change “which” to “that”
- Line 367: Change “has been reported also” to “has also been reported”. Please use “also” in the correct location throughout the entire article
- Line 368: Change “recognizing” to “, which recognizes”
- Line 374: Add “,” before “while”
- Line 384: Add “has” before “demonstrated”
- Line 386: Add “,” before “which”
- Line 392: Add “,” before “suggesting”; change “PAM” to “PAMs”
- Line 401: Please use “Cas9” instead of “cas9” throughout the entire article
- Line 402: Delete “to carryout targeted genome mutagenesis”
- Line 406: Delete “an”
- Line 408-409: Delete “Several designs of the expression cassette have been developed for improved efficiency of CRISPR-Cas9 system.”
- Line 428: Change “class” to “classes”
- Line 465: Add “,” before “these”
- Line 471: Change “spatio-temporal” to “spatiotemporal” throughout the entire article
- Line 477: Add “a” before “Cas9”
- Line 478: Add “,” before “the Pol II Cestrum”
- Line 493: Change “pretty works well” to “works pretty well”
- Line 496: Change “The fact that” to “Since”
- Line 511: Delete “of”; add “a” before “Cas9”
- Line 517: Delete “to result in an”
- Line 539: Change “double CaMV35S minimal promoter and Arabidopsis enhancer” to “a double CaMV35S minimal promoter and an Arabidopsis enhancer”
- Line 597: Change “in worst case” to “in the worst case,”
- Line 607-609: Delete “Efficient simultaneous expression of multiple sgRNAs is crucial for a successful multiplexing strategy using CRISPR-Cas9 system. To date, several approaches have been developed and are currently in use.”
- Line 610: Add “a” before “expression”
- Line 617: Change “Introns are engineered also to carry stacked of multiple sgRNAs with dedicated RNA processing system” to “Introns are also engineered to carry stacked sgRNAs with dedicated RNA processing systems”
- Line 620: Change “complex” to “complexes”
- Line 631: Does “one downstream nucleotide” mean “a downstream sequence”?
- Line 641-642: Please rewrite this sentence
- Line 682: Change “class” to classes
- Line 698: Add “,” before “the sgRNA cassette”
- Line 700: Add “,” before “allowing” throughout the entire article
- Line 707: Change “which” to “that”
- Line 708-709: Please rewrite this sentence.
- Line 712: Change “On the basis of” to “Based on”
- Line 720: Add “the” before “mature mRNA”
- Line 722: Add “,” before “suggesting”
- Line 724: Change “recombining” to “combining”
- Line 727: Add “,” before “breaking”
- Line 733-736: Please break up this sentence to two.
- Line 762: Add “,” after “tools”
- Line 763-764: Change “as well as in revolutionizing” to “and”
- Line 771-772: Delete “Delivering the CRISPR-Cas9 tools is crucial for achieving efficient targeted genome mutation.”
- Line 775-777: Delete “As with conventional plant trangenesis, the delivery of CRISPR-Cas9 reagents in plants is achieved using these transformation methods and have been proven to successfully deliver the editing system in various amendable plant species.”
- Line 786-788: Delete “The increased likelihood of generating transgene modified plants using DNA-based CRISPR-Cas9 system has led to the realization of Cas9-gRNA Ribonucleoproteins (RNPs), a DNA-free version of CRISPR-Cas9 platform.”
- Line 842: Change “of delivering” to “to deliver”
- Line 881: Change “of” to “by”
- Line 907: Change “gene-gene” to “gene”
- Line 946: Add “is” before “still”
- Line 961: Add “,” before “hampering”Line 961: Add “,” before “making”
- Line 966: Change “shorter nor simpler” to “short nor simple”
- Line 984: Change “with non-existent” to “without”
- Line 987: Add “,” before “such as”
- Line 1013: Add “,” before “resulting in”; add “thus” before “inhibiting”
- Line 1043: Change “which” to “that”
- Line 1042-1045: Please reorganize this sentence.
- Line 1062: Change “signal appropriately” to “signals”
- Line 1079-1080: Please rewrite this sentence.
- Line 1102: Add “,” before and after “reacts with H2O2”
- Line 1110: Delete “approach”
- Line 1115: Add “,” before “suggesting”
- Line 1116-1117: Change “both dicot and monocots” to “including both dicots and monocots,”
- Line 1118: Change the subtitle to “Improve CRISPR-Cas9 efficiency by RNA-silencing inhibition and heat stress”
- Line 1124: Delete “undeniably”
- Line 1131: Keep the format of “Arabidopsis” consistent throughout the entire article
- Line 1158-1159: Move “still” after “the application of CRISPR-Cas9 in plants is”
- Line 1160: Add “,” before “are”
- Line 1162-1165: Please simplify this sentence
- Line 1170: Change “resulting to” to “, resulting in”
- Line 1184: Add “delivery” after “biolistic”
- Line 1194: Change “application” to “approach”
- Line 1197: Delete “approach”; change “held” to “hold”
- Line 1198: Add “the” before “extended adoption”
- Line 1201: Change “variety” to “lines”
- Line 1207: Change “with no possibility of” to “without”
- Line 1209: Change “rendered” to “renders”
- Line 1223-1226: Please simplify this sentence
- Line 1230: Delete “a”
- Line 1231: Delete “the duration of”
- Line 1245: Change “extended also” to “also extended”

Author Response
Reviewer 1
We are truly grateful for your valuable efforts in critically reviewing our manuscript. Indeed, your insightful comments and recommendations substantially helped us in improving our manuscript. We have prepared our responses to each point that you have raised. For your convenience, the new line number in the revised manuscript is indicated in red colored font.
- Line 128-129: The information provided here regarding to the classification of the CRISPR systems is not up to data. Please cite the latest classification of CRISPR systems: Makarova, K.S., Wolf, Y.I., Iranzo, J. et al. Evolutionary classification of CRISPR–Cas systems: a burst of class 2 and derived variants. Nat Rev Microbiol 18, 67–83 (2020). https://doi.org/10.1038/s41579-019-0299-x
Response: We have inserted the article in the manuscript as you suggested. Line 128.
- Line 161-162: The single RNA chimera talked here is usually referred as a single guide RNA (sgRNA), while a guide RNA is usually referred as a gRNA. gRNA is also commonly used for other CRISPR systems. But gRNA does not necessarily mean sgRNA specifically used in the Cas9 system. Please keep this consistent and accurate through the entire manuscript.
Response: We have changed the term ‘gRNA’ to ‘sgRNA’ to maintain consistency of the term all throughout the manuscript. Line 161.
- Figure 3(A) is not correct. The crRNAs are first transcribed as a CRISPR array, then hybrid with tracrRNA, followed by the cleavage of Cas9 and RNase III.
Response: The figure has been modified according to the proper stages of CRISPR-Cas9-mediated immunity in bacteria. The caption of the figure was also modified. Line 170.
- Line 340-342: The authors stated that a high GC content is correlated with a high editing efficiency. This statement is not accurate. Editing efficiency will be decreased when the GC content is too high or too low, usually higher than 60% or lower than 40%.
Response: We have modified the statement, back upped with an appropriate reference. Line 337.
- Line 376: Recently, a near-PAMless SpCas9 variant (SpRY) has been engineered with an expanded targeting scope. Although this Cas9 variant has not been demonstrated in plants, it is still a breakthrough in this field and will be soon applied in plants. Therefore, it is worth discussing here with the following citation: Walton, R.T., Christie, K.A., Whittaker, M.N., and Kleinstiver, B.P. (2020). Unconstrained genome targeting with near-PAMless engineered CRISPR-Cas9 variants. Science 368, 290–296.
Response: We have included the details of this study in the revised manuscript. Line 373-384.
- Line 433-435: Although this statement is supported by the following two citations, this statement is too conclusive. It might not absolutely true in other cases or plant species.
Response: We have retained the statement in the manuscript. Aside from the fact that the statement is founded by the findings of reliable studies, we believed that the whole subsection was written in a well-balanced manner. Please refer to line 455-458.
- Line 489: In this paragraph, the authors discussed the advantages of using PoI II promoters. It is worth mentioning that the use of PoI II promoter to drive sgRNA expression requires the sgRNA processing system. Any extra sequences in sgRNA could potentially decrease the editing efficiency or completely abolish its function, although it is less an issue when extra sequences are present at the 3’end of Cas9 sgRNA.
Response: Details regarding the sgRNA processing systems have been added in the text. Line 502-509
- Line 669: The authors stated that the tRNA processing system is precise. This might be a bit overstated, since tRNA cleavage will leave 1-4 nt extra tails at the 3’ end of the sgRNA. This does not affect the sgRNA function in the Cas9 system, but could cause problems when accurate cleavage is required, such as in the Cas12a system.
Response: From the literature that we have gathered, most of the authors described the sgRNA processing activity of the tRNA system as ‘precise’, although extra nucleotides are present in the processed sgRNA. As we have understood from their study, they used the term precise to denote the defined cleavage at the sgRNA-tRNA junction, allowing the liberation of individual sgRNAs from the polycistronic transcript. In this context, we thought that the word precise may not be referring to how precise the sequences of sgRNAs obtained after being excised from the transcript. We apologize for not making this point clear in our manuscript. To avoid confusion, we have used the word “robust” in lieu of the word “precise”. Line 686.
- Line 676-679: Please refer to Figure 5.
Response: Figure 5 was referred. Line 696.
- Line 697-701: Although the use of PolyA or linker sequence instead of a terminator can assure the continuous transcription of sgRNAs, it is also worth mentioning that PolyA serves as an important structure for mRNA to potentially facilitate its exportation, translation, and stability.
Response: Details on the importance of PolyA linker sequence have been added. Line 718-719
- Line 733-738: Please indicate that these speculations are from the original research paper, not from this review paper.
Response: The statements were revised to direct the speculations to the rightful reference. Line 754-757
- Line 873: The authors mentioned earlier that “the cargo capacity of RNA viruses is limited”. It is better to discuss the possibility to deliver both Cas9 and sgRNAs using virus, since Cas9 has a large size.
Response: The point has been added in the text. Line 883-887, 889-892
- Line 884-909: Please add citation for these two paragraphs. Citations usually cannot be carried over to the following paragraphs.
Response: References were added to the said paragraphs. Line 903-926
- Line 949: This sentence is overstated. Genome editing can also be detected using other methods than genomic sequencing, such as enzyme digestion-based methods.
Response: The statement has been modified. We used the term ‘molecular approaches’ to cater other molecular-based approaches for mutant validation. Line 968
- Line 950-951: This sentence is overstated. Whether someone chooses transgene-free approach or not should depends on the objective of the project.
Response: The statement has been omitted.
- Line 952: It is highly recommended to add another figure for this section to show all the detection methods for transgene-free edited plants.
Response: A figure for this section has been added. Figure 7. Line 986
- Line 963-964: This statement is not accurate. Transgenic plants can also be deregulated, but usually require a lengthy and expensive process. Transgene-free edited plants can potentially not be regulated at some levels in certain countries.
Response: We added the word ‘likely’ in the statement to lower down the tone of the claim. Line 981
- Line 1023-1025: The statement is not accurate. The mixtures of bentazon resistant and susceptible plants could also be due to that some plants are not successfully transformed. The resistant plants may contain no transgene at all, but not have silenced transgenes.
Response: Yes, this could be possible. However, the authors of the cited studies used ‘transgene silencing’ to describe the bentazon resistant regenerated plants. In the introductory part of their paper, they mentioned the probability of T-DNA silencing in T0 regenerated plants, which impedes the proper selection of T0 plants carrying the transgene and with the desired gene mutation. In this regard, their method of screening was established to solve this tedious process of selecting T0 plants with active transgenes. They did not regard the resistant T0 plants as unsuccessfully transformed plants. They must have presented other data to support their claim, aside from the insignificant low level transcripts of Cas9 in the resistant plants.
On the other hand, in our manuscript we used the phrase “suggesting a strong possibility” (unedited version) to indicate the transgene silencing phenomenon in the transformed plants, making our statement inconclusive. We believe that our statement only denotes the likelihood of the transgene silencing phenomenon as the cause of the observed resistant phenotype of the regenerated plants. Line 1060-1086
- Line 1037: This study described by Lu et al. 2017 were not able to detect silenced T-DNA. If the T-DNA is silenced in the plant genome, RNAi components can not be expressed and used for herbicide resistance selection. Therefore, edited herbicide resistant plants could potentially contain silenced T-DNA insertion. If this is the case, please reorganize this paragraph and the previous paragraph to avoid misleading.
Response: No, that isn’t the case. The authors only mentioned a possible transgene or T-DNA silencing event after transformation, in T0 regenerated plants. Yes, if the T-DNA is silenced in the genome, herbicide resistant phenotype can be observed due to the lack of active RNAi components targeting the herbicide resistance gene in plants. This phenomenon was expected by the author to occur during the first stage of the screening process – screening T0 plants with the transgene. No edited herbicide resistant plants with silenced T-DNA can be expected from the process, as no genome editing mediated by CRISPR-Cas9 components can occur when T-DNA is silenced. The edited herbicide resistant plants have only been isolated in T1 generation, after successful genetic segregation of T0 sensitive plants. Line 1060-1086
- Line 1152: It is recommended to rewrite this section, since this section is lengthy and repeated many contents from the previous sections. Please keep it short and precise and avoid repeating what have been discussed intensively before.
Response: The summary and concluding remarks section has been rewritten. We have tried to make it concise as possible. Line 1202-1285
- Line 1155: The word “superseded” may be overstated. Although CRISPR is the predominant approach to achieve genome editing, other technologies, such as TALEN, still have some advantages in some circumstances. For instance, TALEN is not limited by PAM sequences. In addition, a better choice for genome editing in organelles is TALEN, since directing protein into organelles is easier than directing RNA.
Response: Yes, that could be true. However, we have retained the word ‘superseded’ as we think this word does not result in an exaggeration of our statement. We have clearly specified in the statement as to why and to what extent the CRISPR-Cas9 system superseded the other known genome editing tools. Line 1204-1206
- Line 1166: It is worth mentioning the development of high-fidelity Cas9 variants to minimize off-target effects.
Response: The point has been added in the text. Line 1214-1217
- Line 1248-1250: This study by Aschenbrenner et al. 2020 mainly focused on limiting off-target effects. This part should be moved to previous sections where off-target effect is discussed.
Response: Details regarding this study have been moved to Cas9 variants subsection. Line 385-389
- Line 1259-1260: This sentence might be overstated based on a single study.
Response: The statement has been omitted.
- Line 1261-1263: Since the topics regarding regulation and public education is not the focus of this review, this is not an appropriate ending for this review.
Response: The statement has been omitted.
- The writing of this review can be greatly improved. It is highly recommended that the authors should let a native speaker editing this article during the revising process. It is important to keep the language concise and precise. Here are some recommendations to improve the writing:
Response: All of the following recommendations have been executed. New line numbers are indicated in red colored font.
- Line 40: Change “highlighted” to “highlights”
- Line 41: Add “is” before “favored”
- Line 43: Add “,” before “which” throughout the entire article.
- Line 46: Add “,” before “which”
- Line 55: Add “,” before “resulting”; change “resulting to” to “resulting in” throughout the entire article
- Line 69: Change “allowed” to “allows”
- Line 73: Change “has shown great advantage and more straightforward” to “has shown great advantages and is more straightforward”
- Line 74: Change “targeted-DSBs” to “targeted”
- Line 77: Add “more” before “versatile”
- Line 78: Change “rate” to “rates”
- Line 79: Change “on” to “about”
- Line 82: Add “,” before “however”. Please do this throughout the entire article
- Line 90-91: Change “straightforward and enhanced genome editing efficiency of the system” to “simple and efficient genome editing systems”
- Line 96-97: Change “involved aversion of phage attachment, blocking DNA entry” to “involve phage attachment aversion, DNA entry blockage” line 95-96
- Line 104: Add “,” before “which” line 103
- Line 118: Add “a” before “CRISPR array” line 117
- Line 119: Change “system shows” to “systems show” line 118
- Line 120: Add “are” before “currently classified” line 118
- Line 143: This sentence “Following the stages of CRISPR-Cas immunity” is not clear, please rewrite it. line 142
- Line 144: Change “the host genome, encoded as CRISPR array” to “the CRISPR array in the host genome” line 143
- Line 151: Change “having” to “with” line 150
- Line 162: Change “made the system simplified” to “simplified the system” line 161
- Line 170: Change “specific DNA sequence with the recognized PAM” to “specific DNA sequences with recognizable PAMs” line 169
- Line 170-171: Delete “, directing the nuclease activity of Cas9 to the target DNA sequence”
- Line 179-180: Change “a fusion construct of crRNA and tracrRNA which is transcribed forming a gRNA” to “a construct of a single guide RNA, which is a fusion of a crRNA and tracrRNA” line 177-178
- Line 180: Add “a” before “complex” line 178
- Line 184: Change “significant array” to “a large number of”; delete “have” before “proven”; delete “functional” line 182
- Line 187: Change “knock out and knock in gene alterations” to gene knockout and knockin” line 185
- Line 188: Add “,” before respectively. Please do this throughout the entire article line 186
- Line 196: Change “to functionally characterized genes” to “characterize gene functions” line 193
- Line 206: Change “high cost and time consuming process” to “their costly and time-consuming processes”; Change “approach” to “approaches” line 203
- Line 207: Change “has” to “have” line 204
- Line 217: Change “time” to “time-” line 214
- Line 220: Add “,” before and after “respectively” line 217
- Line 221: Change “been developed also” to “also been developed”; move “simultaneously” to the end of the sentence line 218-219
- Line 224: Delete “too”
- Line 225: Add “as well” after [57] line 222
- Line 227: Add “,” before “respectively” line 223
- Line 228: Change “in generating” to “to generate” line 225
- Line 253: Change “harbor mutation” to “harboring mutations” line 250
- Line 258: Change “which” to “that” line 255
- Line 259: Change “has been reported also” to “has also been reported” line 256
- Line 264: Add “,” before “thereby” line 261
- Line 265: Change “field drought condition” to “drought conditions” line 262
- Line 293-294: Add “,” before and after “which were previously characterized based on spontaneous mutants or RNAi knockdown lines” line 290-291
- Line 300: Add “the” after “paved”; change “network” to “networks” line 297
- Line 304: Add “the” before “CRISPR” line 301
- Line 307: Change “exerted” to “exerts” line 304
- Line 308: Change “mutagenesis of gene” to “gene mutagenesis” line 305
- Line 311-312: Change “to regard with in employing the technology” to “regarding the application of the technology” line 308-309
- Line 318: Add “the” before “overall” line 315
- Line 327: Change “and should be selected considering that it lies upstream of” to “followed by” line 324
- Line 328: Add “the” before “Cas9” line 325
- Line 329: The word “Consequently” is not used properly here – omitted, line 325
- Line 330-331: Please rewrite this sentence line 327
- Line 335: Move “in silico” after sgRNAs line 331
- Line 341: Change “has been shown also” to “has also been shown” line 337
- Line 355: Change “particular concern specially” to “a particular concern especially” line 351
- Line 356: Change “which” to “that” line 352
- Line 367: Change “has been reported also” to “has also been reported”. Please use “also” in the correct location throughout the entire article line 363
- Line 368: Change “recognizing” to “, which recognizes” line 364
- Line 374: Add “,” before “while” line 370
- Line 384: Add “has” before “demonstrated” line 397
- Line 386: Add “,” before “which” line 399
- Line 392: Add “,” before “suggesting”; change “PAM” to “PAMs” line 405
- Line 401: Please use “Cas9” instead of “cas9” throughout the entire article – we have used the ‘cas9’ to indicate the cas9 gene, especially in the section concerning molecular construct of the vector.
- Line 402: Delete “to carryout targeted genome mutagenesis” deleted
- Line 406: Delete “an” deleted
- Line 408-409: Delete “Several designs of the expression cassette have been developed for improved efficiency of CRISPR-Cas9 system.” deleted
- Line 428: Change “class” to “classes” line 440
- Line 465: Add “,” before “these” line 477
- Line 471: Change “spatio-temporal” to “spatiotemporal” throughout the entire article line 483
- Line 477: Add “a” before “Cas9” line 489
- Line 478: Add “,” before “the Pol II Cestrum” line 490
- Line 493: Change “pretty works well” to “works pretty well” line 513
- Line 496: Change “The fact that” to “Since” line 516
- Line 511: Delete “of”; add “a” before “Cas9” line 531
- Line 517: Delete “to result in an” line 538
- Line 539: Change “double CaMV35S minimal promoter and Arabidopsis enhancer” to “a double CaMV35S minimal promoter and an Arabidopsis enhancer” line 559
- Line 597: Change “in worst case” to “in the worst case,” line 364
- Line 607-609: Delete “Efficient simultaneous expression of multiple sgRNAs is crucial for a successful multiplexing strategy using CRISPR-Cas9 system. To date, several approaches have been developed and are currently in use.” (figure 5 caption) line 606
- Line 610: Add “a” before “expression” line 607
- Line 617: Change “Introns are engineered also to carry stacked of multiple sgRNAs with dedicated RNA processing system” to “Introns are also engineered to carry stacked sgRNAs with dedicated RNA processing systems” line 614
- Line 620: Change “complex” to “complexes” line 617
- Line 631: Does “one downstream nucleotide” mean “a downstream sequence”? line 647-648 . statement has been modified.
- Line 641-642: Please rewrite this sentence line 658-660
- Line 682: Change “class” to classes line 700
- Line 698: Add “,” before “the sgRNA cassette” line 716
- Line 700: Add “,” before “allowing” throughout the entire article line 718
- Line 707: Change “which” to “that” line 726
- Line 708-709: Please rewrite this sentence. Line 728-731
- Line 712: Change “On the basis of” to “Based on” line 731
- Line 720: Add “the” before “mature mRNA” line 739
- Line 722: Add “,” before “suggesting” line 742
- Line 724: Change “recombining” to “combining” line 744
- Line 727: Add “,” before “breaking” line 747
- Line 733-736: Please break up this sentence to two. Line 755-748
- Line 762: Add “,” after “tools” line 782
- Line 763-764: Change “as well as in revolutionizing” to “and” line 783
- Line 771-772: Delete “Delivering the CRISPR-Cas9 tools is crucial for achieving efficient targeted genome mutation.” deleted
- Line 775-777: Delete “As with conventional plant trangenesis, the delivery of CRISPR-Cas9 reagents in plants is achieved using these transformation methods and have been proven to successfully deliver the editing system in various amendable plant species.” deleted from figure 6 caption, line 817
- Line 786-788: Delete “The increased likelihood of generating transgene modified plants using DNA-based CRISPR-Cas9 system has led to the realization of Cas9-gRNA Ribonucleoproteins (RNPs), a DNA-free version of CRISPR-Cas9 platform.” Deleted from figure 6 caption, line 817
- Line 842: Change “of delivering” to “to deliver” line 855
- Line 881: Change “of” to “by” line 901
- Line 907: Change “gene-gene” to “gene” line 927
- Line 946: Add “is” before “still” line 966
- Line 961: Add “,” before “hampering” line 979
- Line 961: Add “,” before “making” line 981
- Line 966: Change “shorter nor simpler” to “short nor simple” line 984
- Line 984: Change “with non-existent” to “without” line 1034
- Line 987: Add “,” before “such as” line 1036
- Line 1013: Add “,” before “resulting in”; add “thus” before “inhibiting” line 1063
- Line 1043: Change “which” to “that” line 1093
- Line 1042-1045: Please reorganize this sentence. line 1093-1094
- Line 1062: Change “signal appropriately” to “signals” line 1112
- Line 1079-1080: Please rewrite this sentence. line 1129-1130
- Line 1102: Add “,” before and after “reacts with H2O2” line 1153
- Line 1110: Delete “approach” line 1161
- Line 1115: Add “,” before “suggesting” line 1167
- Line 1116-1117: Change “both dicot and monocots” to “including both dicots and monocots,” line 1168
- Line 1118: Change the subtitle to “Improve CRISPR-Cas9 efficiency by RNA-silencing inhibition and heat stress” line 1170
- Line 1124: Delete “undeniably” line 1176
- Line 1131: Keep the format of “Arabidopsis” consistent throughout the entire article – executed
- Line 1158-1159: Move “still” after “the application of CRISPR-Cas9 in plants is” line 1210
- Line 1160: Add “,” before “are” line 1211
- Line 1162-1165: Please simplify this sentence line 1213-1215
- Line 1170: Change “resulting to” to “, resulting in” line 1221
- Line 1184: Add “delivery” after “biolistic” (the whole sentence was deleted in the new version of the manuscript)
- Line 1194: Change “application” to “approach” line 1239
- Line 1197: Delete “approach”; change “held” to “hold” line 1242
- Line 1198: Add “the” before “extended adoption” line 1243
- Line 1201: Change “variety” to “lines” line 1246
- Line 1207: Change “with no possibility of” to “without” line 1249
- Line 1209: Change “rendered” to “renders” (the whole sentence was deleted in the new version of the manuscript)
- Line 1223-1226: Please simplify this sentence line 1261-1262
- Line 1230: Delete “a” line 1267
- Line 1231: Delete “the duration of” line 1268
- Line 1245: Change “extended also” to “also extended” line 1281

Reviewer 2 Report
This is an extensive and up-to-date review that significantly contributes to existing literature. It was enjoyable and educational to read.
The English should be revised by a native, as there are numerous mistakes throughout the manuscript, but especially in the Introduction and Concluding Remarks. See lines: 36-40, 73, 107, 184, 187...1165, 1182, 1230, 1234...
Check the accuracy of numbered references, for example Feng et al., is not [115] as cited (line 449).
Avoid excessive and informal description as in lines 305-306 "A myriad..."
Have you adapted Figure 3 from another source? It looks familiar. If so, please say so.
Author Response
Reviewer 2
This is an extensive and up-to-date review that significantly contributes to existing literature. It was enjoyable and educational to read.
Response: Thank you so much for quoting our manuscript as enjoyable and educational to read. We also thank you for your insightful comments and suggestions. Indicated in here are our responses to your concerns. The new line number in the revised manuscript is indicated in red colored font.
- The English should be revised by a native, as there are numerous mistakes throughout the manuscript, but especially in the Introduction and Concluding Remarks. See lines: 36-40, 73, 107, 184, 187...1165, 1182, 1230, 1234...
Response: The introduction and the summary and concluding remarks sections have been revised, especially on the technical aspect of the English grammar.
- Check the accuracy of numbered references, for example Feng et al., is not [115] as cited (line 449).
Response: We apologize for the wrong reference given. The correct reference has been cited in the revised manuscript. Line 461
- Avoid excessive and informal description as in lines 305-306 "A myriad..."
Response: Informal description has been changed. Line 302
- Have you adapted Figure 3 from another source? It looks familiar. If so, please say so.
Response: Figure 3 was inspired from the paper of Sander and Joung, 2014. We apologize for not indicating where the figure was adapted in our first version of the manuscript. However, in the revised manuscript, figure 3 has been intensively modified following the comments of the other reviewer. With this, we believed that the modifications made fully differentiated our figure from where it was initially inspired. Line 171

Reviewer 3 Report
This review introduces the fundamentals of CRISPR-Cas9 system and summarizes its application in plant genome editing, meanwhile discusses the current challenges and emerging advancements and optimization approaches in this biotechnology in plant. It is comprehensive and scientific.
Author Response
Reviewer 3
This review introduces the fundamentals of CRISPR-Cas9 system and summarizes its application
in plant genome editing, meanwhile discusses the current challenges and emerging advancements
and optimization approaches in this biotechnology in plant. It is comprehensive and scientific.
Response: We sincerely thank you for taking your time in reading our manuscript. We are happy
to know that you find our paper comprehensive and scientific in dealing with the topic.
Round 2
Reviewer 1 Report
The authors have put a great effort to revise the manuscript and improve the writing. The current version of the manuscript is much better and more accurate than the previous one.